



# Zooplankton faecal pellet transfer through the meso- and bathypelagic layers in the Southern Ocean in spring

Anna Belcher[1,2], Clara Manno[3], Pete Ward[3], Stephanie Henson[1], Richard Sanders[1], Geraint Tarling[3]

[1]National Oceanography Centre, Southampton, SO14 3ZH, UK
[2]University of Southampton, Southampton, SO14 3ZH, UK
[3] British Antarctic Survey, Cambridge, CB3 0ET, UK

*Correspondence to*: Anna Belcher (A.Belcher@noc.soton.ac.uk)

**Abstract.** The faecal pellets (FP) of zooplankton can be important vehicles for the transfer of particulate organic carbon (POC) to the deep ocean, often making large contributions to carbon sequestration. However, the routes by which these FP reach the deep ocean have yet to be fully resolved. We address this by comparing estimates of FP production to measurements of FP size, shape and number in the upper mesopelagic (175-205 m), using Marine Snow Catchers, and in the bathypelagic, using sediment traps (1,500-2,000 m). The study is focussed on the Scotia Sea, which contains some of the most productive regions in the Southern Ocean, where epipelagic FP production is likely to be high. We found that, although the size distribution of zooplankton suggests that high numbers of small FP are produced in the epipelagic, small FP are rare in the deeper layers, implying that they are not transferred efficiently to depth. Consequently, small FP make only a minor contribution to FP fluxes in the meso- and bathypelagic, particularly in terms of carbon. The dominant FP in the upper mesopelagic were cylindrical and elliptical, while ovoid FP were dominant in the bathypelagic. The change in FP morphology, as well as size distribution, points to the repacking of surface FP in the mesopelagic and in situ production in the lower meso- and bathypelagic, augmented by inputs of FP via zooplankton vertical migrations. The flux of carbon to the deeper layers within the Southern Ocean is therefore strongly modulated by meso- and bathypelagic zooplankton, meaning that the community structure in these zones has a major impact on the efficiency of FP transfer to depth.

## 1 Introduction

The biological carbon pump (BCP) from the atmosphere to the deep ocean is an important process by which carbon can be sequestered for millennia or longer (Volk and Hoffert, 1985). About 10% of surface ocean primary production sinks out (is exported) of the surface ocean, with the remainder being remineralised in situ. However, only a small fraction of this material (<10%) reaches the deep ocean (Sarmiento and Gruber, 2006), with most of it being respired by grazers or bacteria (Azam et al., 1983) in the upper mesopelagic (Martin et al., 1987). Thus close to 10% of surface primary production is stored in the interior, a process which keeps atmospheric $CO_2$ around 200 ppm lower than preindustrial levels (Parekh et al., 2006). Small changes in the BCP, such as a change in the depth at which sinking material is remineralised can result in large



changes to the climate system; a global increase of 24 m in the depth at with 63% of sinking carbon is respired could
decrease atmospheric $CO_2$ by 10-27 ppm (Kwon et al., 2009). For this reason, the nature of particles occurring at different
depths is important to understand.

The repackaging of slow-sinking individual phytoplankton cells into fast-sinking faecal pellets (FP) can promote efficient
export of POC out of the euphotic zone (Hamm et al., 2001). The contribution of FP to bathypelagic particle fluxes can be
large (>90%) (Carroll et al., 1998; Manno et al., 2015; Wilson et al., 2013), providing direct evidence of the importance of
zooplankton FP to the transport of carbon to the deep ocean. However, surface produced FP can also undergo intense
reworking and fragmentation in the euphotic and upper mesopelagic zones (González et al., 1994b; Riser et al., 2001;
Wexels Riser et al., 2007), through processes such as coprophagy (ingestion of FP), coprorhexy (fragmentation of FP),
microbial remineralisation and physical aggregation and disaggregation (Lampitt et al., 1990; Poulsen and Iversen, 2008;
Turner, 2015; Wilson et al., 2008). Thus, FP can also provide a source of nutrition for other zooplankton and bacterial
communities en route to the deep ocean (Miquel et al., 2015; Riser et al., 2001). The complexity of these interacting factors
results in a wide range of estimates (<1->100% (Turner, 2015)) of the contribution FP make to POC flux (%FPC), which is
typically measured using sediment traps (Dagg et al., 2003; Fowler et al., 1991; Gleiber et al., 2012; Manno et al., 2015;
Suzuki et al., 2001; Wassmann et al., 2000; Wilson et al., 2013).

Differences in FP shape, composition and density, as well as varying depths of production (through zooplankton species
residing at different depths and also vertical migration (VM)) will greatly influence the magnitude of FP associated POC that
reaches the deep ocean (Atkinson et al., 2012; Steinberg et al., 2000; Wallace et al., 2013; Wilson et al., 2008). Both diel and
seasonal migrations of zooplankton can directly transport carbon out of the euphotic zone to the mesopelagic, bypassing the
region of rapid remineralisation (Jónasdóttir et al., 2015; Kobari et al., 2008; Steinberg et al., 2000). Different zooplankton
feeding strategies will also influence the effect that their vertical migrations have on POC export (Wallace et al., 2013).

The direct sinking of zooplankton FP can provide an efficient vehicle for the sequestration of carbon in the deep ocean. For
example, direct sedimentation of FP from large salp blooms in the upper ocean can result in huge depositions on the sea floor
at depths of ~4000 m due to their high sinking velocities (Smith, Jr. et al., 2014). Additionally, the swarming behaviour of
krill can result in *en masse* sinking of FP, which can overload recycling zooplankton grazers and be efficiently transferred
through the upper ocean (Clarke et al., 1988). Alternatively, FP may arrive in the deep ocean via a FP 'cascade' effect
(Bodungen et al., 1987; Urrere and Knauer, 1981), being constantly reworked and transformed with depth. The fact that FP
have been observed in the deep ocean highlights the important role they play in carbon sequestration, however knowledge of
the route by which these FP reach the deep ocean is not yet clear. There is a need for comparisons between the composition
and characteristics of sinking FP just below the euphotic zone and in the deep ocean to improve our understanding of both



the origin of faecal material reaching the deep ocean and how it is potentially modified by meso- and bathypelagic zooplankton.

Here we use Marine Snow Catchers and deep ocean sediment traps in the Scotia Sea, within the Southern Ocean, to collect intact sinking FP in the upper mesopelagic and bathypelagic respectively, and use these data to compare the characteristics of mesopelagic and bathypelagic FP. We compare zooplankton abundances in the upper 200 m with FP fluxes in both the upper mesopelagic and bathypelagic in order to understand the processes controlling the fate of FP produced in the epipelagic. We use these data to determine whether FP arriving in sediment traps in the deep ocean are a result of a direct detrital rain from the surface, or are produced in the mesopelagic via the grazing and repackaging of this material by deep zooplankton populations. We focus in particular on copepod FP as copepods are the numerically dominant zooplankton in our study region, typically comprising >90% of total zooplankton (Ward et al., 2012). Zooplankton FP can make a large contribution to fluxes of FP in the meso- and bathypelagic of the Scotia Sea (e.g. Belcher et al., 2016b; Cavan et al., 2015; Manno et al., 2015). In this region, the transfer of FP through the mesopelagic (as well as the mechanisms controlling their transfer) is therefore a key determinant of the efficiency of the BCP.

## 2 Methods

### 2.1 Study site

Sediment traps have been deployed for a number of years at two sites, P2 and P3 (Fig. 1), upstream and downstream of South Georgia (at -55.248 °N, -41.265 °E and -52.812 °N, -39.972 °E respectively) in the Scotia Sea in the Southern Ocean (Manno et al., 2015). Samples were collected from Marine Snow Catcher (MSC) and zooplankton abundance data were collected during cruises in austral spring 2013 (JR291) and 2014 (JR304) aboard the *RRS James Clark Ross* (Table 1). Sediment trap data were obtained from traps deployed in 2012 and 2013 at P2 and P3, at depths of 1,500 m and 2,000 m respectively. The P3 trap (2,000 m depth) was deployed in May 2013 on cruise JR287, and P2 (1,500 m depth) deployed on 8[th] December 2012 on cruise JR280, herein defined as D1. Both traps were recovered in December 2013 on cruise JR291 aboard the *R.R.S. James Clark Ross*. In addition the P2 mooring was redeployed on 7[th] Dec 2013 and recovered on 28[th] November 2014 during cruise JR304, herein defined as D2. Samples from the spring period (October to January) were analysed for comparison with MSC deployments. Mean current velocities at both sites are <10 m s[-1] (Whitehouse et al., 2012) suggesting effects of lateral advection are minimal and as such they are not considered in this study.



**2.2 Mesozooplankton collection**

**2.2.1 Net sampling**

Mesozooplankton were collected using a motion-compensating Bongo net (61 cm mouth diameter, 2.8 m long, 200 µm mesh). The net was equipped with solid cod ends, deployed to 200 m and hauled vertically to the surface at 0.22 m s$^{-1}$. Samples were preserved in 4% formalin (w/v) in seawater before being identified to species/taxa using a binocular microscope and staged where appropriate. At least 500 individuals were counted per sample. Counts were converted into ind. m$^{-2}$ (0-200 m) based on the area of the Bongo net mouth and the depth of deployment. A total of five deployments were carried out during JR291 and two during JR304. Average abundances for each species/taxa were calculated by averaging all the deployments (from both cruises) at each site. Antarctic krill (*Euphausia superba*) and other large euphausiids were occasionally caught in the Bongo nets, but the Bongo net does not accurately quantify their abundance due to their patchy distribution and net avoidance capabilities. Large euphausiid abundances were therefore not considered, so zooplankton abundances in this study reflect mesozooplankton abundances. In particular, copepod species were overwhelmingly dominant in terms of abundance at our study site, typically >90% of total zooplankton abundance (Ward et al., 2012).

**2.2.2 Prediction of faecal pellet size distribution in epipelagic layers**

We predicted the size distribution of FP in the epipelagic layers by using the size distribution of the copepod community assessed via prosome length (PL, mm) (Ward et al. 2012, their table A1) and the known relationship between copepod size and the volume of their FP (FPV, µm$^3$) (Mauchline, 1998; Stamieszkin et al., 2015).

$$\log_{10} FPV = \theta \log_{10}(PL) + \eta \tag{1}$$

We take mean values of θ and η of 5.4 and 2.58 from Stamieszkin et al. (2015) derived from literature values of FPV and PL. Using measured copepod abundances, we then calculated the size distribution of FP produced by our population of copepods. We compared the percent abundance in each size class, making the assumption that all copepods were egesting FP at the same rate (see Discussion). As the zooplankton net tows are integrated from the surface to 200 mm, there is a slight overlap with the MSC samples, however as the bulk of zooplankton are found in the upper 100 m (Ward et al., 2014), these net samples are largely representative of the epipelagic layer and we refer to it as such for simplicity. Non-copepod zooplankton (~10 % mesozooplankton abundance) were not considered in this calculation and represents a background error in this approach.





**2.3 Faecal pellet collection**

**2.3.1 Marine Snow Catcher deployments**

Marine Snow Catchers (MSC) were deployed in the upper mesopelagic, defined here as 110 m below the base of the mixed layer depth (MLD) identified from vertical profiles of the water column taken prior to MSC deployments using a Conductivity-Temperature-Depth (CTD) unit (Seabird 9Plus with SBE32 carousel). MSC are large (95 L) PVC closing water bottles, designed to minimise turbulence so particles are more likely to remain intact (Belcher et al., 2016a, 2016b; Cavan et al., 2015; Riley et al., 2012). Once at the appropriate depth, MSC were closed via a mechanical release mechanism, before recovering and leaving on deck for a settling period (2 hours). Following settling, they were drained and particles that sank fast enough to reach the bottom collector tray ("fast sinking" particles (Riley et al., 2012)) were removed in the tray and stored at 2-4°C for further analysis. Particles reaching the bottom of the tray that were visible by eye (>0.15 mm diameter) were picked from the tray using a wide bore pipette. Given the MSC height of 1.53 m, particles originating at the top of the MSC are required to sink at a minimum rate of 18.4 m d$^{-1}$ to reach the base of the MSC. However considering measurements of FP sinking velocity in the Southern Ocean (27 m d$^{-1}$ to 1218 m d$^{-1}$ (Atkinson et al., 2012; Belcher et al., 2016b; Cavan et al., 2015), this is likely sufficient to capture sinking FP.

**2.3.2 Sediment trap deployments**

Sediment traps (ST) were deployed in the bathypelagic (1500 m to 2000 m). They consisted of a plastic funnel with a baffle at the top (0.5 m$^2$ surface area), and a narrow opening at the bottom, through which particles fall into 1 L sampling cups (McClane, PARFLUX Mark 78H-21). The traps were programmed so sampling cups would rotate after 14 to 31 days, with shorter periods set to coincide with expected periods of high productivity. Prior to deployment each cup was filled with a preservative solution of sodium chloride buffered 0.01% Mercuric Chloride. Upon recovery, samples were photographed and the pH recorded. Swimmers, defined as zooplankton that were alive and intact on entering the trap, were picked out using tweezers and removed from the sample. Each sample was then split into a number of equal aliquots (determined by the amount of material in the sample) using a rotary splitter McClane Wet Sample Divider (WSD-10). Here we focus on ST trap samples in November and December (austral spring) to match MSC and zooplankton net deployments.

**2.4 Faecal pellet analysis**

All FP were photographed using an Olympus SZX16 microscope. FP were classified visually as round, ovoid and cylindrical using light microscopy. All FP in each category collected in the MSC were counted, and their length and width measured using ImageJ. For each ST sample, the dimensions of 10-50 FP of each class were measured. FP volumes were calculated for round, ovoid and cylindrical pellets using the formula for a sphere, ellipsoid and cylinder respectively. Equivalent spherical diameters (ESD) were also calculated. We compare FP volume rather than FP number to avoid bias due to possible fragmentation (Wexels Riser et al., 2010). The carbon contents of FP were calculated based on conversion factors of 0.035,




0.052 and 0.030 mg C mm$^{-3}$ for round, ovoid and cylindrical FP respectively based on measurements made on FP collected
from the ST in spring-early autumn (Manno et al., 2015).

Without faecal production experiments of isolated species, it is difficult to ascertain the exact origin of FP collected in the
MSC and ST. Previous studies (González and Smetacek, 1994; González, 1992; González et al., 1994a; Martens, 1978;
Wilson et al., 2008; Yoon et al., 2001) suggest that ovoid/ellipsoidal pellets originate from copepods, pteropods and
larvaceans, cylindrical from krill and copepods and spherical pellets from amphipods, small copepods and crustacean
nauplii.

### 2.5 Faecal pellet sinking velocities and fluxes

Sinking velocities (*w*) of a sample of FP collected in MSC were measured on board on both cruises. During JR291, sinking
velocities were measured in a graduated glass cylinder in a temperature controlled laboratory (2°C). For each FP, the sinking
velocity was calculated from the average of the time taken to sink past two marked distances (10 cm apart), with the starting
point more than 10 cm from the water surface. During JR304, sinking velocities were measured in a temperature controlled
(at 4°C) flow chamber system (Ploug and Jorgensen, 1999), suspending FP in an upward flow and taking the average of
three measurements. Only FP larger than 0.15 mm ESD (i.e. those visible by eye) could be measured. No significant
differences were found between sinking velocities measured during JR291 and JR304 by these two different methods
(Student's t-test, *p*=0.2).
The median sinking velocity of measured FP for each MSC was utilised to calculate the sinking FP flux (*FPF*).

$$FPF\ (n\ FP\ m^{-2}d^{-1}) = \frac{nFP}{A} \times \frac{w}{h} \qquad (2)$$

Here, *nFP* is the total number of FP collected at the base of the MSC (excluding krill FP), *A* the area of the MSC opening
based on inner MSC diameter, and *h* the height of the snow catcher (1.53 m).
For sediment trap samples, FP fluxes were calculated as follows:

$$FPF\ (n\ FP\ m^{-2}d^{-1}) = nFP/(A/d), \qquad (3)$$

where *d* is the number of days that the trap was open (15 days) and *A* is the area of the sediment trap (0.5 m$^2$).

### 2.6 Faecal pellet comparisons

FP collected in the ST and MSC were compared in terms of the number of FP in each morphological type as well as in terms
of carbon. As the absolute number of FP was vastly different between MSC and ST samples due to attenuation with depth,




we compared the percentage abundance and carbon across the size distribution of all FP from measured FP volumes. As only
an average FP size for each morphological type (rather than for all individual FP) was measured for samples from the ST
deployments D1 and D2, we make use of historical sediment trap data (Manno et al., 2015) at the same sites from 2009 and
2010 (herein referred to as H2009 and H2010). The size of all FP in each sample-split were measured in the study of Manno
et al. (2015) allowing us to compare size distributions of MSC and ST collected FP. Manno et al. (2015) also categorised FP
into ovoid, cylindrical and round, with an additional category of elliptical. We combine cylindrical and elliptical categories
due to their similar morphology and to allow comparison with our MSC data. Although this introduces uncertainty in terms
of inter-annual variability between 2009-2010 (full sediment trap data) and 2013-2014 (Marine Snow Catcher data),
consistency in the FP types and percentages in each category between years (Fig. S2) provides confidence in the use of these
historical data. Numbers of large cylindrical FP, probably originating from large euphausiids, were removed from counts
given the large potential bias in the quantification of these organisms in the net samples. Again we take only the spring data
(November and December).

## 2.7 Statistics

In order to estimate error uncertainty, we take the standard error of our measurements, i.e. multiple Bongo net tows for
zooplankton, multiple MSC deployments for mesopelagic FP, and multiple ST deployments for bathypelagic FP. We
compare zooplankton size distributions using a Kolmogorov-Smirnov test. FP size distributions (in terms of % abundance)
are also compared using an Anderson-Darling k-sample test as this test is more sensitive to differences in the tails and
differences in shift, scale and symmetry when means are similar (Engmann and Cousineau, 2011). All statistics were carried
out in RStudio (version 0.98.1091; R development core team, 2014).

## 3. RESULTS

### 3.1 Zooplankton community and faecal pellet production

On average, total zooplankton abundances and species compositions were similar at P2 and P3 (Fig. 2), with small
microcopepod species *Oithona similis*, *Oncaea sp.* and *Ctenocalanus sp.* outnumbering the main large calanoid copepod
species (*Rhincalanus gigas, Calanoides acutus, Calanus similimus, C. propinquus, Euchaeta spp.,* and *Metridia spp*) (Table
S1, Fig. 2). The number of zooplankton with PL < 2 mm is similar at P2 and P3 (ratio P3:P2 of 1.1), but the abundance of
larger copepods (4-7 mm PL) at P3 was almost double that of P2 (ratio P3:P2 of 1.8) (Fig. S1).
The predicted size distribution of egested FP from our mesozooplankton copepod community highlights that most FP
egested in the epipelagic would be in the smallest size category <0.001 mm$^3$ (97.6 ± 20.3% and 97.0 ± 4.0% at P2 and P3
respectively) with low contributions (<2%) from each of the larger FP size categories (Fig. 3a). The high standard error of
FP <0.001 mm$^3$ at P2 is in part due to very high abundances of *Oithona similis* during one deployment. Removing this net





from the average gives 97.8±13.7% FP<0.001 mm$^3$. The predicted size distributions of FP at P2 and P3 were not
significantly different ($p$>0.5, Mann-Whitney U-test, Kolmogorov-Smirnov test, and Anderson-Darling k-sample test).

## 3.2 Sinking faecal pellets

Sinking faecal pellets collected in MSC (upper mesopelagic) and ST (bathypelagic) are described in terms of size and shape
to assess changes between these two layers.

### 3.2.1 Faecal pellet shape

The morphologies of FP captured in the MSC at P2 were heterogeneous (Fig. 4, Fig. 5a), with cylindrical/elliptical FP, and
round FP making up similarly high percent contributions to the total number of FP. Conversely, a single morphology
dominated in the P3 MSC samples which were cylindrical FP of <0.005 mm$^3$ (Fig. 5c).

All morphological classes found in the upper mesopelagic (MSC samples) were also present in the bathypelagic (ST
samples, Fig. 4). However, the dominant type of FP changed between these two layers (Fig. 5). Ovoid FP made only low
contributions (< 8.3% and <1.4% at P2 and P3 respectively) to total FP abundance in the MSC samples but were the
dominant type in most size categories in the ST samples (up to 25.2% and 13.1% at P2 and P3 respectively, Fig. 5).

### 3.2.2 Faecal pellet size

The predicted FP size distributions of pellets produced in the epipelagic by the net caught zooplankton community were
significantly different to those observed in the upper mesopelagic (MSC samples) at both P2 and P3 (Kolmogorov-Smirnov
test, D=0.58 (P2), D=0.67 (P3), DF=11, $p$<0.01). Comparison of Fig. 3a and b reveals that there was a reduced dominance of
the smallest FP (0-0.001 mm$^3$) from >96 ± <20% to <18 ± <5% between the two layers at both sites.

A further loss in the smaller FP size categories is apparent between the upper mesopelagic MSC samples and the
bathypelagic ST samples (Fig. 3c). FP <0.003 mm$^3$ in volume decreased from 35.5 ± 13.4% to 5.0 ± 0.4% at P2 and from
52.3 ± 6.7% to 14.0 ± 5.7% at P3. Based on size alone, the FP community appears to have become less diverse in the
bathypelagic layer, with most FP (>80 %) occupying a narrower size range in the ST samples, (0.003-0.01 mm$^3$) compared
to the MSC samples (0.001-0.02 mm$^3$). FP size distributions in the MSC and ST were not however significantly different at
either P2 or P3 (Anderson-Darling k-sample test, T.AD=1.3, DF=11, $p$=0.2 and T.AD=0.43, DF=11, $p$=0.9 at P2 and P3
respectively). Re-running the test for only FP size categories <0.003 m$^3$ highlights a significant difference in the %FP
abundance in the smaller size categories between the MSC and ST ($p$=0.03 at both P2 and P3).





### 3.3 Faecal pellet carbon


Although small FP were numerically dominant in the MSC, comparison of Fig. 5 and Fig. 6 reveals higher contributions of
the larger FP size classes to total FP carbon (FPC). This is not unexpected as larger FP contain a larger amount of carbon.
FPC data highlight the importance of the loss of large FP to the carbon sinking through the water column. Although
abundances of small FP greatly reduced with depth, this does not represent such a large change in terms of carbon.

### 3.4 Faecal pellet sinking velocities and fluxes


Sinking velocities of FP collected in the MSC ranged from 61 to 950 m d$^{-1}$ at P2 and 24 to 370 m d$^{-1}$ at P3 reflecting the
range in FP shapes and sizes. Generally small FP had lower sinking velocities than larger FP. During cruise JR291, we
measured FP sinking rates of 65-120 m d$^{-1}$ for FP <0.002 mm$^3$, and 118-207 m d$^{-1}$ for FP >0.02 mm$^3$. During cruise JR304,
FP sinking rates were 47-51 m d$^{-1}$ and 36-270 m d$^{-1}$ for FP <0.002 mm$^3$ and FP >0.02 mm$^3$ respectively.

At P3, the flux of cylindrical and elliptical FP in the MSC was an order of magnitude higher than fluxes of round or ovoid
FP whereas, at P2, cylindrical and elliptical FP were the dominant FP type, but fluxes of round FP were of a similar
magnitude (Table 2). FP fluxes in the ST were dominated by ovoid FP at both sites (Table 2).

## 4. DISCUSSION


In this study we compare predicted size distributions of FP produced by the zooplankton (mainly copepod) community in the
epipelagic, to those of sinking FP in the upper mesopelagic (from MSC) and the bathypelagic (from ST) in order to
determine the fate of FP sinking through the mesopelagic and assess the importance of deep dwelling zooplankton on the
efficiency of the BCP in the Southern Ocean.

### 4.1 Changes in faecal pellet with depth: upper mesopelagic


Our data suggest that small FP are not transferred efficiently from the epipelagic to the meso- and bathypelagic, and hence
make a small contribution to FP fluxes at depth, particularly in terms of carbon. Comparison of estimated copepod FP
production with measurements of sinking FP in the upper mesopelagic (from MSC) gives an indication of the degree of
retention in that layer. The community at both P2 and P3 was dominated by microcopepod species which, based on their
size, produce small FP which are expected to sink more slowly than large FP (Komar et al., 1981; Small et al., 1979;
Stamieszkin et al., 2015). Agreeing with the data presented here, small FP (<0.002 mm$^3$) are predicted to have a sinking
velocity three times slower than larger FP (>0.02 mm$^3$) based on the empirical relationship of Small et al. (1979) for copepod
FP.



The longer residence time of small FP in the upper ocean (due to their slower sinking velocities) means they are exposed to
remineralisation processes such as coprophagous feeding, fragmentation and microbial remineralisation, for a longer period
of time. This type of retention filter and low export efficiency of small FP has been observed in a number of oceanographic
environments (e.g. Dagg et al., 2003; Riser et al., 2001; Viitasalo et al., 1999). Wexels Riser et al. (2010) made observations
over the upper 200 m of a Norwegian fjord, finding that large FP produced by *Calanus finmarchicus* contributed
disproportionately to vertical flux despite large numbers of small FP produced by *Oithona similis,* agreeing well with the
loss of small FP that we observed in the Scotia Sea.

It is important to acknowledge here, that although the 200 µm mesh used in this study is commonly used in zooplankton
surveys, this leads to an underestimation of the smaller zooplankton size classes present in the epipelagic. Ward et al., (2012)
found that a 53 µm mesh caught 5.87 times more zooplankton than a 200 µm net in the upper mesopelagic of the northern
Scotia Sea in spring. However, in this study an underestimation of the small zooplankton size classes serves to reinforce the
fact that small FP dominate the flux of FP out of the epipelagic and are largely attenuated as they pass through the
mesopelagic.

Comparison of freshly egested FP size distributions with the size distributions of FP sinking through the mesopelagic relies
here on the assumption that different species within the copepod community had the same rates of egestion. FP production
varies with species, as well as factors such as season and food availability; the range in FP production rates between different
copepod species across a number of high latitude studies is 2-48 FP ind.d$^{-1}$ (Dagg et al., 2003; Daly, 1997; Roy et al., 2000;
Thibault et al., 1999; Urban-Rich et al., 1999). However, as the estimated abundance of egested FP in the smallest size
category (0-0.001 mm$^3$) is between 60-250 times greater than the next largest category, the smallest FP are still likely to
dominate the FP community even if egestion rates are varied within reasonable bounds. Therefore, despite our assumptions
regarding rates of egestion, our conclusion of rapid attenuation of these small FP in the upper mesopelagic remains valid.
**4.2 Changes in faecal pellet with depth: meso- to bathypelagic**
Our data reveal a change in FP size, shape and abundance between the upper mesopelagic and bathypelagic of the Scotia Sea
suggesting in situ FP production by deeper dwelling zooplankton. The occurrence of intact and fresh FP in deep sediment
traps in the Southern Ocean (e.g. Accornero et al., 2003; Manno et al., 2015) may therefore be a result of an indirect,
cascade-like transfer through the mesopelagic as they are reprocessed by different zooplankton communities (Miquel et al.,
2015; Urrere and Knauer, 1981).

Urrere and Knauer (1981) deployed free-floating traps off the Monterey Peninsula in California. They observed a decrease in
numerical FP fluxes in the upper 500 m, but FP fluxes increased by a factor of 2.7 from 500 m to 1500 m. This increase was
largely due to elliptical FP, suggesting the presence of deep resident (or overwintering) zooplankton populations (Urrere and



Knauer, 1981). The authors conclude that organic material reaches the deep ocean (supporting deep resident zooplankton
populations) through in situ repackaging of detritus and via heterotrophy as well as inputs from migrating populations,
emulating the "ladder of migrations" first proposed by Vinogradov (1962). More recently, Miquel et al. (2015) deployed
drifting sediment traps in the upper 210 m of the Beaufort Sea, observing increases in elliptical FP with depth and decreases
in cylindrical FP. They explain this by the presence of omnivorous and carnivorous zooplankton in the mesopelagic, whose
primary food sources are the vertical flux of organic matter and other organisms. In agreement with our observations, Suzuki
et al. (2003) observed large declines in cylindrical FP between sediment traps deployed at 537 and 796 m in the marginal ice
zone of Antarctica, and increases in elliptical FP over the same depth range. They suggest that coprophagous feeding and
new FP production can explain some of the loss of cylindrical FP, with fragmentation into small sinking particles explaining
the rest. As different zooplankton species produce different shape of FP, a change in FP shape suggests a change in
zooplankton community structure.

At both P2 and P3 we saw an increase in the contribution of ovoid FP to the total number of FP between the upper
mesopelagic (MSC samples) and bathypelagic (ST samples), increasing by factors of 4.5 and 8.5 at P2 and P3 respectively.
This suggests that there is either an input of ovoid FP at depth, or that cylindrical-elliptical and round FP are preferentially
remineralised in the mesopelagic. We made both size and shape measurements of FP in the upper mesopelagic and
bathypelagic, allowing us to discern if there is indeed production of new ovoid FP at depth. At both P2 and P3, we observed
size classes of ovoid FP in the ST (0.003-0.008 mm$^3$) that were not present in the MSC, which rules out selective
remineralisation. Furthermore, the intact shape of ovoid FP in the ST argues against fragmentation as a cause of this change
in size distribution. In agreement with Manno et al. (2015), we observed that ovoid FP in the ST showed fewer signs of
fragmentation and were more intact than cylindrical or elliptical FP at both P2 and P3. Estimates of FPC in ST samples
indicates that these ovoid FP also make a large contribution to the flux of POC and, as such, their production at depth
represents a mechanism for long term storage of carbon in the ocean. Hence, we conclude that FP fluxes to depth are
augmented by FP produced in situ at depth.

We can estimate the size class of zooplankton producing the FP we find at depth based on the FP size class and Equation 1.
We estimate that zooplankton of PL 2.6-3.8 mm and 2.6-3.2 mm could have produced the FP we observed in the ST, based
on dominant size classes of FP of 0.003-0.008 mm$^3$ and 0.003-0.005 mm$^3$ at P3 and P2 respectively. Of the species within
these size classes recorded in the Bongo net tows at P2 and P3, *Calanoides acutus IV* and *Metridia gerlachei* adults were the
most abundant and may be responsible for the flux of these FP to the ST. *C.acutus* is a known seasonal migrator in the
region, occurring in the upper 200 m in summer but residing deeper (~200 -600 m) in spring (Ward et al., 2012). *Metridia*
*spp.* are also known migrators (Ward and Shreeve, 1999; Ward et al., 1995, 2006), found to be one of the more abundant
species in the 500-1000 m depth range based on *Discovery Investigations* to the west of the Drake Passage (Ward et al.,
2014). Ward et al. (2014) find the most abundant species in this depth range to be *Oncaea spp.*, *Oithona frigada* and



*Microcalanus pygmaeus*, all of which are too small (≤0.5 mm PL) to produce the larger FP that were dominant in the ST.
Similar to the situation in the epipelagic and upper mesopelagic, we suggest that although small species are more abundant,
they produce small FP which sink slowly and are rapidly remineralised. It is likely that it is the less abundant larger
carnivores and recyclers in the lower mesopelagic that are contributing more to the flux of carbon to the deep ocean through
the production of large FP, agreeing with the modelling study of Stamieszkin et al., (2015). Calanoid copepod families,
*Aetideidae*, *Heterorhabdidae, Metridinidae* and *Euchaetidae* are also common in the mesopelagic of the Scotia Sea and
surrounding area (Laakmann et al., 2009; Ward and Shreeve, 1999; Ward et al., 1995), and are of an appropriate size (as
adults or other copepodite stages) to produce the larger FP that were dominant in the ST. Although we can only speculate as
to the possible producers of FP in the ST, it is clear that appropriately sized zooplankton are sufficiently abundant in the
mesopelagic to influence the flux of FP to the ST.
**4.3 Role of meso- and bathypelagic zooplankton**
Our data suggest that zooplankton residing below the euphotic layer repackage sinking detritus and produce FP which are
able to pass through the lower mesopelagic and be collected in ST in the bathypelagic. Observations made at P2 and P3 in
autumn show that, during the night, the highest zooplankton abundances are in the upper 125 m (C.Liszka pers. comm.).
However corresponding daytime surface abundances are typically lower which may be partially explained by certain species
that migrate vertically in the water column (C.Liszka pers. comm.). We suggest that diel vertical migrators may contribute to
the relatively fresh FP we found at depth. A modelling study by Wallace et al. (2013) suggests that FP penetrate deeper in
the water column when there is zooplankton vertical migration, with the deepest FP production occurring when zooplankton
undertake diel vertical migrations rather than foray type feeding (multiple ascents and descents during a day). Resident
zooplankton populations were observed below 150 m depth, with a peak at 375-500 m, most notably at P3 (C.Liszka
pers.comm.), suggesting that the non-migrating, or seasonally or ontogenetically migrating, community are also important at
our study site and could repackage organic material in the upper mesopelagic, producing some of the intact FP which we
observed in our ST.

The abundance of zooplankton typically declines rapidly over the upper 1000 m of the water column (Ward and Shreeve,
1999; Ward et al., 1995, 2014), suggesting that any new FP production below the depth of our MSC samples is likely to take
place in the upper to mid mesopelagic where zooplankton abundances are higher. Although zooplankton are more
concentrated in the epipelagic, total depth integrated zooplankton abundances in the 250-2000 m horizon (extending
abundances recorded at 750-1000 m down to 2000 m) in the Antarctic Zone (to the west of our study site) is about three
quarters (0.74) of the number of zooplankton in the top 250 m (Ward et al., 2014). Therefore it is likely that there is still
substantial production of FP in the lower mesopelagic, and FP produced here are subject to remineralisation processes over a
shorter distance so are more likely to reach the deep ocean intact.



Despite the similarities in copepod abundances at P2 and P3, the numbers of FP collected at P3 were an order of magnitude
higher than at P2. Surface phytoplankton productivity at P3 is typically much higher than at P2, with large blooms occurring
in most years (Borrione and Schlitzer, 2013; Korb et al., 2008, 2012). This may in part explain higher FP fluxes at the P3
site, as in good feeding conditions (such as those measured during JR304 (Belcher et al., 2016b)) FP production rates have
been shown to be higher (Besiktepe and Dam, 2002; Butler and Dam, 1994). The zooplankton community structure may also
affect the fate of FP in the mesopelagic. Previous studies have found relationships between POC export and the presence of
microcopepod species, suggesting that low POC export may be attributed to coprophagy and/or coprorhexy (Suzuki et al.,
2003; Svensen and Nejstgaard, 2003). More recently, several studies have proposed that the main role of small zooplankton
species may be to fragment FP rather than ingest them (Iversen and Poulsen, 2007; Poulsen and Kiørboe, 2005; Reigstad et
al., 2005). Regardless of the mechanism, previous studies agree that high microcopepod abundances can lead to increased FP
retention. The increased abundance of small copepods (compared to larger calanoids) at P2 (Figure 2) may result in greater
losses of FP in the epi- and mesopelagic, resulting in lower numbers of FP captured in our MSC and ST at P2. Indeed, we
see higher attenuation of FP fluxes at P2 than P3 between our measurement depths (Table 2).

The flux of FP reaching the deep ocean therefore depends not only on surface production, but also on the meso- and
bathypelagic zooplankton populations and the balance between FP retention and FP production. For instance, if the deep
zooplankton community at P3 are larger in size than those at P2, this could explain the larger size of FP observed in the ST
at P3 as well as contributing to higher numbers of FP here due to increased sinking velocities of larger FP (Komar et al.,
1981; Small et al., 1979; Stamieszkin et al., 2015). Although our data implies that in situ FP production in the mesopelagic
accounted for additional fluxes of FP to the bathypelagic at both P2 and P3, the potential for further working and
fragmentation of FP produced in the mesopelagic means we are not able to quantify this deep FP production. We therefore
cannot determine whether higher FP fluxes at P3 are due primarily to reduced attenuation or to increased FP production at
depth, however at least in the upper mesopelagic (mixed layer depth-200 m) FP attenuation is higher at P2 than P3 (Belcher
et al., 2016b). We cannot rule out that a combination of both is occurring.

We present here a comparison of FP size, shape and abundance in the upper mesopelagic and lower bathypelagic allowing us
to verify previous hypotheses of in situ FP production and vertical migrations augmenting the flux of FP to depth in the
Southern Ocean (Accornero et al., 2003; Manno et al., 2015; Suzuki et al., 2003). We find that the occurrence of intact FP in
deep ST can be explained by both vertical migrations of zooplankton, and repackaging and in situ FP production by meso-
and bathypelagic zooplankton populations (Fig. 7). The route by which the FP are transferred to depth is a key control on the
amount of carbon reaching this depth. Taking an integrated surface production of 1 g C m$^{-2}$ d$^{-1}$ (based on measurements by
Korb et al. (2012) to the northwest of South Georgia), and assuming that FP reaching the deep ocean via vertical migration
are only assimilated (with efficiency of 66% (Anderson and Tang, 2010; Head, 1992)) once (left panel Fig. 7, Case A), we
calculate that up to 340 mg C m$^{-2}$ d$^{-1}$ could reach the depth of migration. In comparison, if we assume FP undergoing



repackaging in the mesopelagic are assimilated twice over the same depth range as the vertical migration (right panel, Fig. 7, Case B), up to 115 mg C m$^{-2}$ d$^{-1}$ could reach the same depth. These estimates are within the range of estimates of POC flux made in the upper mesopelagic at P2 and P3 (Belcher et al., 2016b), but are over an order of magnitude higher than POC fluxes measured in the ST (Manno et al., 2015), implying that material reaching the ST may have been repackaged many times. The exact difference in carbon transfer between these two routes (Case A and B) will depend on the number of repacking steps, specific assimilation efficiencies of the repackaging copepods as well as degree of microbial remineralisation occurring during FP sinking between repackaging cycles. Regardless of the feeding mode of these mesopelagic zooplankton communities (detritivory, omnivory or carnivory), production of FP at depth via both the aforementioned scenarios supports the transfer of intact FP to the deep ocean, supporting the sequestration of carbon on long timescales. There is therefore a need to link meso- and bathypelagic zooplankton communities (particularly the larger size classes) to carbon fluxes within global biogeochemical models by refining the contribution of different zooplankton size classes to carbon fluxes via their differential FP production rates and sinking speed.

**Acknowledgements**

We would like to thank the crew, officers and scientists aboard the *R.R.S. James Clark Ross* during research cruises JR291 and JR304. Particular thanks to Elena Ceballos Romero, Fred le Moigne, Andy Richardson, and Manon Duret for their invaluable help with marine snow catcher deployments. Thanks to Cecilia Liszka for providing information on the deep mesozooplankton community at our study site. Fieldwork was supported by a NERC AFI Collaborative Gearing Scheme grant to Stephanie Henson. Geraint A. Tarling and Clara Manno were supported by the Ocean Ecosystems programme at British Antarctic Survey.

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



## Tables

**Table 1: Details of marine snow catcher (MSC) deployments during cruises JR291 and JR304 to the Scotia Sea**

| Cruise | Site | Latitude | Longitude | Date | Time (GMT) | Depth of MSC (m) |
|--------|------|----------|-----------|------|-----------|------------------|
| JR291 | P2 | -55.192 | -41.342 | 02/12/2013 | 23:45 | 176 |
| | P2 | -55.196 | -41.332 | 03/12/2013 | 15:54 | 204 |
| | P2 | -55.259 | -41.295 | 07/12/2013 | 15:07 | 203 |
| | P3 | -52.769 | -40.155 | 13/12/2013 | 13:49 | 205 |
| | P3 | -52.769 | -40.154 | 14/12/2013 | 06:33 | 180 |
| | | | | | | |
| JR304 | P3 | -52.8116 | -39.9727 | 12/12/2014 | 22:40 | 176 |
| | P3 | -52.8118 | -39.9726 | 13/12/2014 | 22:47 | 183 |

**Table 2: FP fluxes (±SE, nFP m$^{-2}$ d$^{-1}$) of ovoid, cylindrical and elliptical (Cyl+Ell), and round FP at P2 and P3 as measured in Marine Snow Catchers (MSC) and sediment traps (ST) in the Scotia Sea in spring.**

| | P3 | | | | P2 | | | |
|--------|-------|-----------|-------|-------|-------|-----------|-------|-------|
| | **Ovoid** | **Cyl + Ell** | **Round** | **Total** | **Ovoid** | **Cyl + Ell** | **Round** | **Total** |
| **MSC** | 13,416 | 190,716 | 32,172 | 236,304 | 6,309 | 21,128 | 14,596 | 89,850 |
| | (±8,207) | (±51,623) | (±15,239) | (±63,079) | (±2,698) | (±1,328) | (±1,124) | (±11,922) |
| **ST** | 11,226 | 7,406 | 4,668 | 23,300 | 640 | 238 | 175 | 1,052 |
| | (±706) | (±1,274) | (±14) | (±1,994) | (±33) | (±82) | (±37) | (±152) |
| **MSC/ST** | 1.2 | 25.8 | 6.9 | 10.1 | 9.9 | 88.9 | 83.5 | 39.9 |



**Figures and Figure Legends**

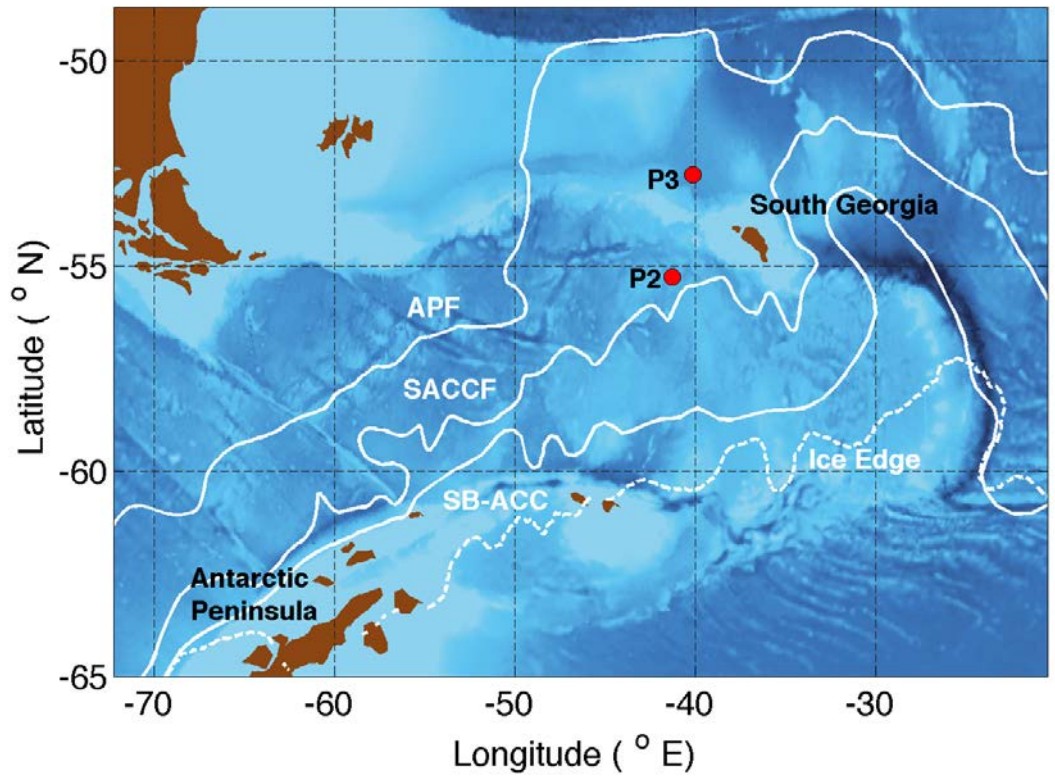


**Figure 1: Stations sampled in the Scotia Sea. White lines indicate average frontal positions. APF=Antarctic Polar Font (Orsi et al.,**
**1995), SACCF = Southern Antarctic Circumpolar Current Front (Thorpe et al., 2002), SB-ACC-=Southern Boundary - Antarctic**
**Circumpolar Current (Orsi et al., 1995). White dotted lines indicates the position of the ice edge on 3[rd] Dec 2013 (OSTIA Sea Ice**
**satellite data).**



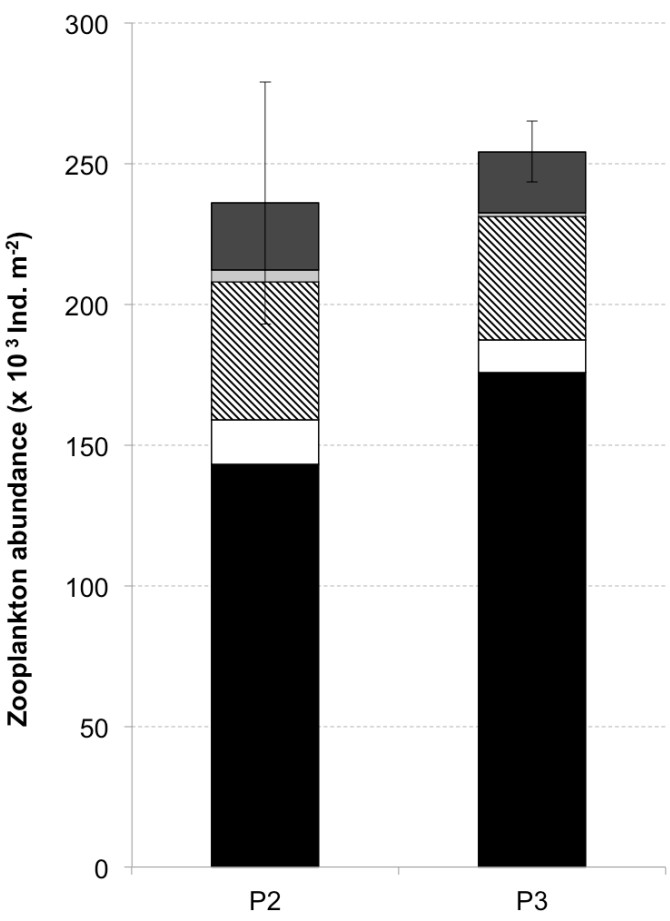

**Figure 2: Average zooplankton abundances (x 10 $^3$ Ind. m$^{-2}$ (0-200m)) measured in the Scotia Sea in December 2013 and 2014 using a 200 µm mesh. Small microcopepods (black), large calanoids (white), other copepods (striped), small euphausiids (light grey), other zooplankton (dark grey) (see text for full details on groups). Error bars show ±SE of total zooplankton abundance based on multiple Bongo net tows at each site.**



621

**Figure 3: Faecal pellet size distributions for P2 (left) and P3 (right) in the Scotia Sea. The percent (%) abundance of faecal pellets in each size class (volume, mm³) is presented for; a) estimated egested faecal pellet size distributions based on mesozooplankton abundances (200 µm mesh), b) faecal pellets measured in marine snow catchers (MSC) at MLD+110 m averages (±SE), and c) faecal pellets in sediment traps (ST). Krill faecal pellets have been removed. Note the uneven faecal pellet volume size classes, and log scale on the Y axis for a.**



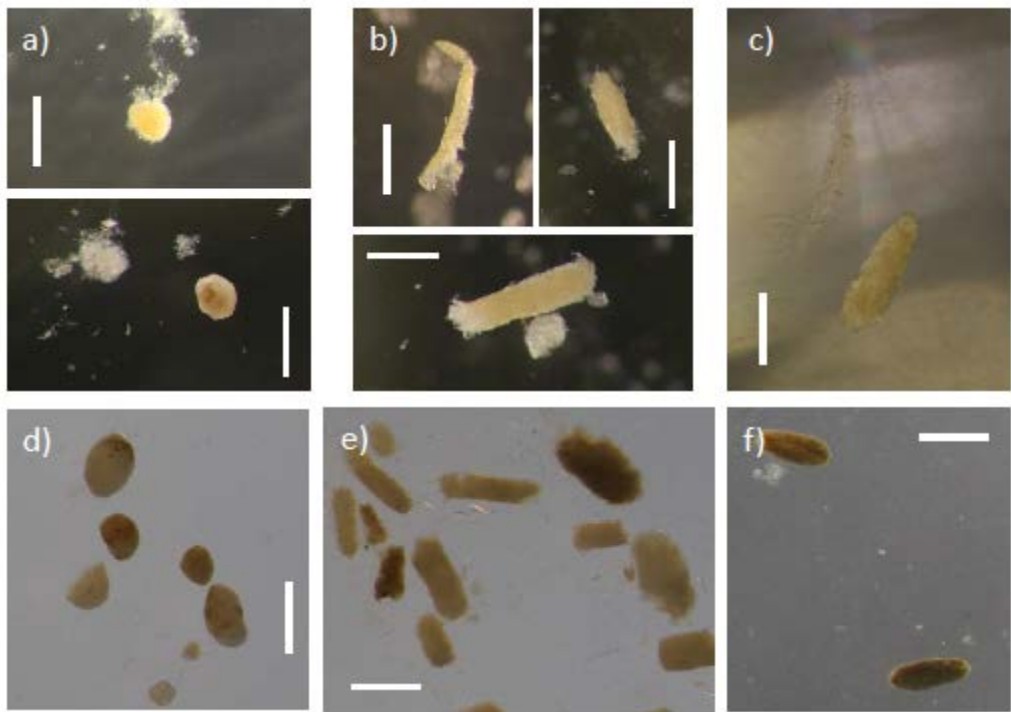


**Figure 4: Light microscopy photographs of faecal pellets collected from Marine Snow Catchers (A-C) and sediment traps (D-F). The different morphological classes are illustrated; a)+d)) round, b)+e) cylindrical, c)+f) ovoid. Scale bar = 0.5 mm.**




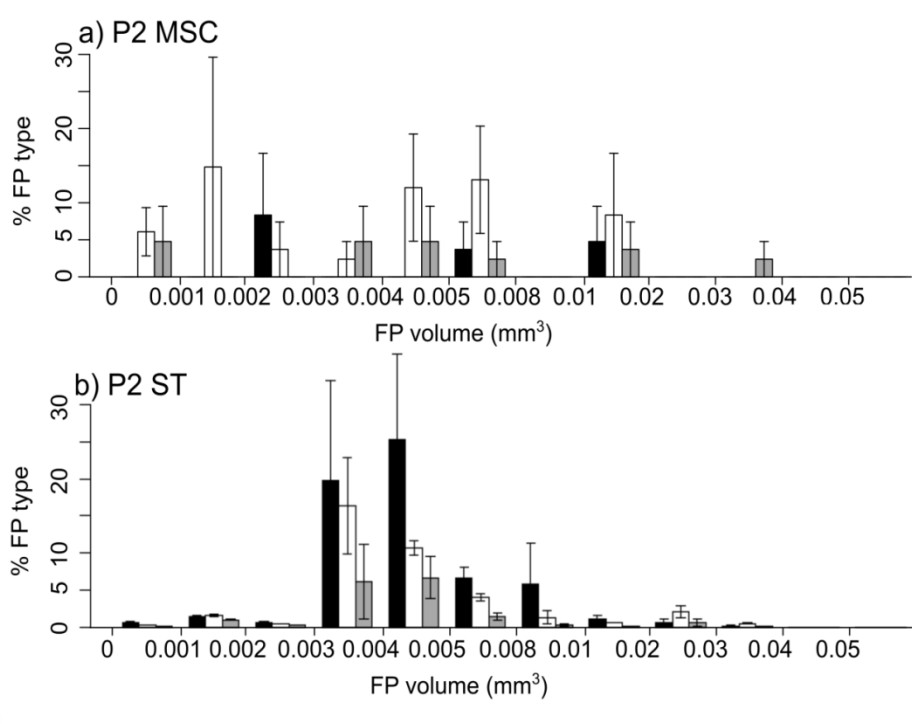

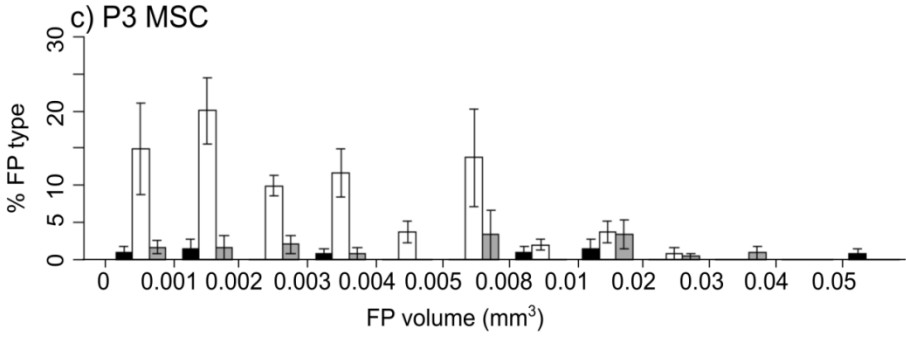

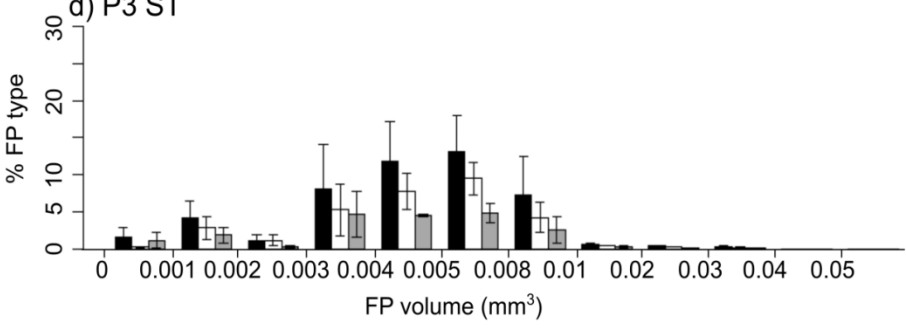

**Figure 5: Percent (%) contribution of each pellet type to total faecal pellet abundance, ovoid (black), cylindrical and elliptical**
**(white) and round (grey). FP from a) P2 Marine Snow Catcher, b) P2 sediment trap, c) P3 Marine Snow Catcher, d) P3 sediment**
**trap. Krill faecal pellets have been removed. Note the uneven faecal pellet volume size classes.**




**Figure 6: Percent (%) contribution of each pellet type to total faecal pellet carbon, ovoid (black), cylindrical and elliptical (white) and round (grey). FP from a) P2 Marine Snow Catcher, b) P2 sediment trap, c) P3 Marine Snow Catcher, d) P3 sediment trap. Krill faecal pellets have been removed. Note the uneven faecal pellet volume size classes.**





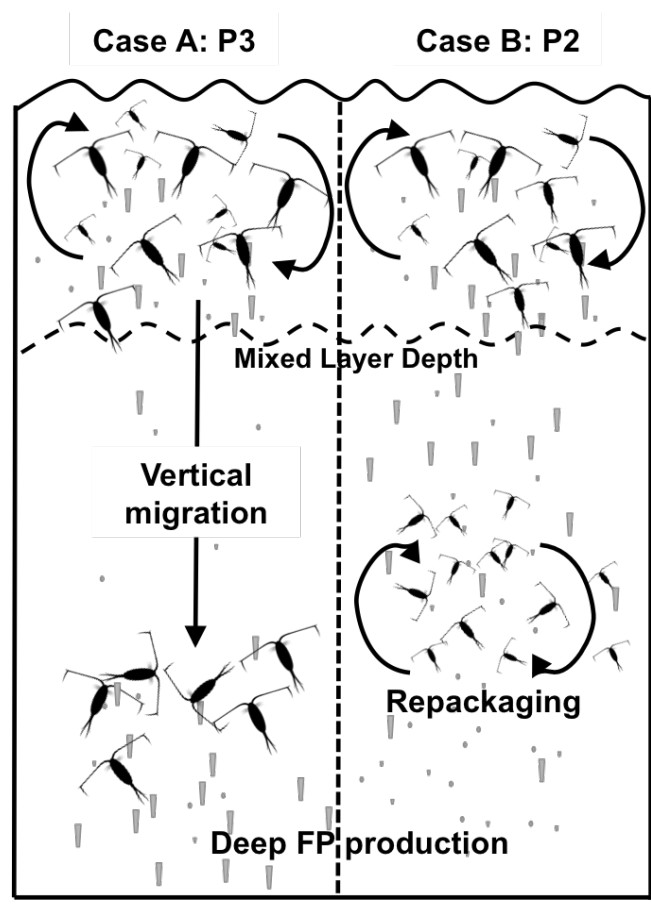

639

**Figure 7: Schematic to illustrate the dominant mechanisms of deep FP production that are suggested to be occurring at our study sites P2 (right) and P3 (left). In Case A, intact FP reach the deep ocean via vertical migration of zooplankton, whereas in Case B, FP at depth are due to in situ repackaging of sinking detritus by deep dwelling zooplankton.**