# Peer review of "Zooplankton faecal pellet transfer through the meso- and bathypelagic layers in the Southern Ocean in spring"

_Biogeosciences, 2016_

## Referee Comment (RC1) · Anonymous Referee #1 · 9 Jan 2017

"Zooplankton fecal pellet transfer through the meso- and bathypelagic layers in the Southern Ocean in spring" by Belcher, A., Manno, C., Ward, P., Henson, S., Sanders, R., Tarling, G.

Belcher et al. estimated in their manuscript the fecal pellet (FP) production of copepods in the epipelagic layer and measured then the FP size, shape and number of FP in the meso- and bathypelagic to investigate the FP transfer to these depth layers. They found that smaller, cylindrical and elliptical shaped FP dominated in the upper mesopelagic in the Marine Snow Catcher, while larger, ovoid shaped FP prevailed in the sediment traps in the bathypelagic. The authors claim that the larger, ovoid shaped FP are an important vehicle for the POC transport to depths and that they are either a result of

repackaging of surface produced FP in the upper mesoplagic or in situ produced FP in the lower meso- and bathypelagic, augmented by vertically migrating zooplankton.

General comments: The manuscript presents an interesting approach of field data, measured data (sinking velocity) and estimation of data, which were not assessed during the cruise. The arguments presented are to a large extend well thought through and the manuscript is well written. In my opinion, the work absolutely deserves publication after some minor revisions.

The authors claim in the title that they investigate the zooplankton fecal pellet transfer, but apart from Fig. 2 and L116-118 the focus is on copepod (fecal pellets) only. I therefore ask the authors to revise their manuscript, because the title and abstract right now imply a study of the whole zooplankton community (also e.g. krill), but this is not the case.

Further, the authors argue (amongst others in the abstract L19) that vertical migration of zooplankton augmented the in situ production of FP in the lower mesopelagic and bathypelagic layer. However, the manuscript does not contain any data on vertical migration (VM), and despite it is likely that VM is an important factor, I ask the authors to revise their manuscript and be more cautious in the conclusions (as nicely done in 356-358) – or presenting addition data on VM.

Similarly, I think that the schematic illustration (Fig.7) draws to sharp conclusions. In my opinion, the data presented do not provide evidence that only VM caused the change in FP size and shape at P3, while only repackaging caused it at P2. Also, the conclusion at the end of the discussion implies that there several repackaging steps may have occurred, but this is not included in Fig. 7. I generally like conceptual figures a lot, but they need to be concise, because they otherwise give easily a wrong impression. Therefore, I ask the authors to revise the conclusion and the schematic illustration to make sure that it matches the data presented. Adding depth terms, that are used in the manuscript (meso-/bathypelagic) as well as the MSC and ST depth deployment may

further help to give a good schematic picture of the situation experienced.

Specific comments:

Title: See general comment

Introduction: 25-28: The authors claim that 10% of primary production sink out and that <10 % reach the deep ocean. That's fine for me, but I have problem understanding then the following sentence "close to 10% of surface primary production is stored in the interior." To my knowledge benthic processes may also use some of the primary production and I am a bit confused by the many "10%". Please revise the language to make it easier understandable.

30-32: Similar to the example above, also this sentence contains lots of numbers, which confuse me somewhat (btw. what is an increase in depth? 24 m deeper?). Consider revising the sentence in a way, which makes it easier to understand, but keep it, because it leads the reader nicely to the reason of the present study.

74-77: Move further up in this paragraph? E.g. L69?

Methods: 82: I would recommend including few short sentences about the study site to introduce it to your reader. Which currents to you find? Do resident deep zooplankton populations drift with it? Potentially drifting patterns could also help to explain the stronger attenuation at P2?

83ff: I struggled a bit with the different locations abbreviations (D1/ D2/ P2/ P3) and would suggest sticking to one label (e.g. P2/ P3 as also marked in the map).

85-89: Consider moving this information to the paragraph "sediment trap deployment". I think this would be a more natural place to have it

93ff: Where have the nets been taken?

111: literature values – did you have similar zooplankton species as in Stamieszkin et al (2015) that you can assume that the estimated made are meaningful?

144: How did you do the classification of the FP? I totally understand that you get a feeling for it, but was it more a subjectively choice (eye-balling) or did you e.g. take pictures and use roundness/ ratio of the minor/major axis? This is more a question out of curiosity, but I think that some standards should be established to make FP classification as objective as possible.

Results: 251-253: This sentence confuses me... Please revise the language.

Discussion: 255: mainly copepod? See general comments

277-282: Good and important point!

362-365: Please revise language to make easier understandable

379: "the increased abundance of small copepods at P2": I am unsure what you mean by the term "increased abundance" (increased compared to what?) and Fig. 2 shows that there were more small copepods at P3 compared to P2. Thus, I do not understand the argument made in line 379-380.

383-387: Do you have any data that you have a larger deep ZP community at P3 than at P2? Otherwise the argumentation is a bit thin.

387-392: Can you please make your arguments a bit clearer? I have problems understanding what you mean here.

392: Consider replacing the sentence by something like: "Most likely a combination of both mechanisms takes place."

396ff: Please be a bit more cautions in your conclusions as you were previously in the discussion (line 351).

398-412: Please revise to make it easier for your reader to follow (e.g. 400: deep ocean – what do you mean by that; 402: depth of migration – how deep it that?) and also revise your Fig. 7 to interlink text and figure better).

Figures and Tables: Figure 1: You do not mention the different fronts in the text. Is there any reason to include them here? In my opinion, depth lines could be more interesting – perhaps that could also help to explain the stronger attenuation of FP at P2?

Figure 7: See general comment

Technical corrections: 205: change to "<2 mm" to be consistent (no space after the <)

205: change "is similar" to "was similar"

192: past tense? Change to "we took into account"?

Figure 2: Other zooplankton (see text for full information…) – I could not find that information

Figure 3/ 5 / 6: Unsure about the labelling of the x-axis. Why do you have the step from 0.005 to 0.008?

Table 2: Minor aspect, but is there a reason, why you put P2 on the right side of the table and P3 on the left side? Intuitively, I would do expect it the other way round (as in Fig. 2)

―――――――――――――――

---

## Referee Comment (RC2) · Anonymous Referee #2 · 6 Feb 2017

General comments The manuscript by Belcher et al. compares estimates of copepod faecal pellet production in the epipelagic with those of faecal pellet abundances in the meso- and bathypelagial derived from Marine Snow Catcher (MSC) and sediment traps. The study was conducted in the Scotia Sea, Southern Ocean. Based on faecal pellet morphology and abundance, the main conclusion of this study is that small faecal pellets are in high abundance in the epipelagic but do not contribute much to export fluxes. Instead, repackaging of faecal pellets and de novo production take place in the meso- and bathypelagic. The manuscript is well written. The findings are in accordance with expectations derived from earlier studies on faecal pellet export in the ocean.

Specific comments I suggest that the authors provide more information on their methods and measurements to allow for better evaluation of the data. I also suggest a more critical discussion on the comparability of data on faecal pellet fluxes derived from net tows, MSC and sediment traps. 1) Lines 89 and 90; here, an estimate for a mean current velocity at the study site of <10 m s-1 is given and Whitehouse et al. (2012) are cited to suggest that lateral advection can be neglected. It is not clear where the current velocity was measured and for what processes it can be neglected. I assume that a current meter was used in the cited study to estimate the trapping efficiency of the sediment traps, but this information should be given. Is the current velocity in the epi- and mesopelagic in the Scotia Sea in the same range? Does one have to consider lateral advection of faecal pellets in the water column? This is important to estimate how well samples from a MSC and deep traps can be compared. 2) 140ff.; please indicate how many splits were analyzed. Were the splits analyzed separately so that a sampling error can be given? How many pellets per split, or in total, were counted? So far only relative abundances are given in the manuscript and supplemental information. The authors may consider providing a table with absolute counts in the supplemental information. Please give this information also for faecal pellets determined in the MSC samples. 3) 159 ff.: The quality of the faecal pellet sinking velocity measurement and therewith of the faecal pellet fluxes cannot be evaluated. The authors state that they used two different approaches to determine faecal pellet sinking velocity and that there were no significant differences between the methods. But how reliable are the obtained sinking velocities? The range of sinking velocities given in line 246 is rather large (24-950 m d-1). The ranges given in lines 248-249 are much smaller. How do these numbers compare? What was the variability of sinking velocity within each approach? What was the variability within each size class? Since these data are used to calculate the faecal pellet flux (FPF), the original data should be given in the manuscript and their accuracy assessed critically. Please add more information. 4) The authors conclude that small FP that sink more slowly are not transferred efficiently to depth as they are subject to remineralization and coprophagy for a much longer period of time than fast sinking large particles. This is very well comprehensible. However, I would like

to see a more critical discussion of the comparability of data from net tows, MSC and sediment traps, which takes into account the spatio-temporal variability in that region, the three-dimensional flow field and the current velocities. In addition to biogenic loss process, slowly sinking fecal pellets found in the MSC of stations P 2 and P3 may not be represented well in the sediment traps at 1500 and 2000m depth, because the traps collect particles from a much wider area (e.g. Waniek et al. 2000) and integrate over a longer, and different, time period.

---

## Author Comment (AC1) · 22 Feb 2017

**Response to reviewer #1**

AC: Thank you for taking the time to read our manuscript and for the suggestions, please find our responses below. Line numbers refer to the marked up version of the manuscript which has also been uploaded.

**Anonymous Referee #1**

"Zooplankton fecal pellet transfer through the meso- and bathypelagic layers in the Southern Ocean in spring" by Belcher, A., Manno, C., Ward, P., Henson, S., Sanders, R., Tarling, G.

Belcher et al. estimated in their manuscript the fecal pellet (FP) production of copepods in the epipelagic layer and measured then the FP size, shape and number of FP in the meso- and bathypelagic to investigate the FP transfer to these depth layers. They found that smaller, cylindrical and elliptical shaped FP dominated in the upper mesopelagic in the Marine Snow Catcher, while larger, ovoid shaped FP prevailed in the sediment traps in the bathypelagic. The authors claim that the larger, ovoid shaped FP are an important vehicle for the POC transport to depths and that they are either a result of repackaging of surface produced FP in the upper mesoplagic or in situ produced FP in the lower meso- and bathypelagic, augmented by vertically migrating zooplankton.

General comments: The manuscript presents an interesting approach of field data, measured data (sinking velocity) and estimation of data, which were not assessed during the cruise. The arguments presented are to a large extend well thought through and the manuscript is well written. In my opinion, the work absolutely deserves publication after some minor revisions.

The authors claim in the title that they investigate the zooplankton fecal pellet transfer, but apart from Fig. 2 and L116-118 the focus is on copepod (fecal pellets) only. I therefore ask the authors to revise their manuscript, because the title and abstract right now imply a study of the whole zooplankton community (also e.g. krill), but this is not the case.

AC: Thank you for highlighting this. We have amended the abstract and title of the manuscript to correctly match that our study focuses on the copepod community.

Further, the authors argue (amongst others in the abstract L19) that vertical migration of zooplankton augmented the in situ production of FP in the lower mesopelagic and bathypelagic layer. However, the manuscript does not contain any data on vertical migration (VM), and despite it is likely that VM is an important factor, I ask the authors to revise their manuscript and be more cautious in the conclusions (as nicely done in 356-358) – or presenting addition data on VM.

AC: We appreciate your comments here and agree that we do not have data on vertical migration and hence need to make sure that our conclusions reflect this. We have amended the text to make sure this is the case. In particular we have revised:

The abstract (lines 18-20)

"The change in FP morphology, as well as size distribution, points to the repacking of surface FP in the mesopelagic and in situ production in the lower meso- and bathypelagic, which may be augmented by inputs of FP via zooplankton vertical migrations."

Section 4.3 (lines 512-516)

"Our comparison of FP size, shape and abundance in the upper mesopelagic and lower bathypelagic agrees with previous

hypotheses (Accornero et al., 2003; Manno et al., 2015; Suzuki et al., 2003), that in situ FP production augments the flux of FP to depth in the Southern Ocean. We find that the occurrence of intact FP in deep ST could be explained by both vertical migrations of zooplankton, and repackaging and in situ FP production by meso- and bathypelagic zooplankton populations (Fig. 7)."

Similarly, I think that the schematic illustration (Fig.7) draws to sharp conclusions. In my opinion, the data presented do not provide evidence that only VM caused the change in FP size and shape at P3, while only repackaging caused it at P2. Also, the conclusion at the end of the discussion implies that there several repackaging steps may have occurred, but this is not included in Fig. 7. I generally like conceptual figures a lot, but they need to be concise, because they otherwise give easily a wrong impression. Therefore, I ask the authors to revise the conclusion and the schematic illustration to make sure that it matches the data presented. Adding depth terms, that are used in the manuscript (meso-/bathypelagic) as well as the MSC and ST depth deployment may further help to give a good schematic picture of the situation experienced.

AC: Thank you for your suggestions to improve Figure 7. We have amended the text (lines 514-572, see specific comments below) to make our arguments clearer and to avoid statements beyond the limits of our dataset. We have shown two repackaging steps in Figure 7 to align with our descriptions in section 4.3. Additionally, we have labelled the epipelagic and mesopelagic/upper bathypelagic layers to make the schematic easier to understand. We no longer present repackaging and vertical migration as two cases that match P2 and P3, but rather refer to them as scenario 1 and scenario 2, stating in the figure legend that likely a complex combination of both scenarios is occurring. As our comparisons of the two scenarios are applicable to any specific depth within the meso- and bathypelagic, and in order not to over complicate the schematic, we do not add the specific deployment depths of the MSC and ST.

Specific comments:

Title: See general comment
AC: Amended to: Copepod faecal pellet transfer through the meso- and bathypelagic layers in the Southern Ocean in spring

Introduction: 25-28: The authors claim that 10% of primary production sink out and that <10 % reach the deep ocean. That's fine for me, but I have problem understanding then the following sentence "close to 10% of surface primary production is stored in the interior." To my knowledge benthic processes may also use some of the primary production and I am a bit confused by the many "10%". Please revise the language to make it easier understandable.

AC: Sentences have been amended as follows (lines 26-30):

"About 10% of surface ocean primary production sinks out (is exported) of the surface ocean, with the remainder being remineralised in situ. However, only a small fraction of this material (<10%) reaches the deep ocean (Sarmiento and Gruber, 2006), with most of it being respired by grazers or bacteria (Azam et al., 1983) in the upper mesopelagic (Martin et al., 1987). Nevertheless, it is estimated that the BCP keeps atmospheric $CO_2$ around 200 ppm lower than preindustrial levels (Parekh et al., 2006)."

30-32: Similar to the example above, also this sentence contains lots of numbers, which confuse me somewhat (btw. what is an increase in depth? 24 m deeper?). Consider revising the sentence in a way, which makes it easier to understand, but keep it, because it leads the reader nicely to the reason of the present study.

AC: Sentence altered for clarity (lines 30-37):

"Small changes in the BCP, such as a change in the depth at which sinking material is remineralised can result in large changes to the climate system; if the depth at with 63% of sinking carbon is respired is increased by 24 m globally, this could decrease atmospheric $CO_2$ by 10-27 ppm (Kwon et al., 2009)."

74-77: Move further up in this paragraph? E.g. L69?

AC: Paragraph reordered as suggested.

Methods: 82: I would recommend including few short sentences about the study site to introduce it to your reader. Which currents

to you find? Do resident deep zooplankton populations drift with it? Potentially drifting patterns could also help to explain the stronger attenuation at P2?

AC: We have added in a few sentences to describe the study site in more detail, noting the differences between the P2 and P3 sites. See section 2.1 lines 88-96 (see below).

"The Scotia Sea is mainly located in the eastward flowing Antarctic Circumpolar Current (ACC), which is split by a number of frontal systems including the Southern Antarctic Circumpolar Front (SACCF, Fig. 1). The complex circulation patterns and variability in frontal systems shapes the Scotia Sea ecosystem (Murphy et al., 2007). P3 and P2 are located downstream and upstream of South Georgia respectively, leading to marked differences in community structure with large rapidly sinking diatoms likely to be more prevalent in the iron fertilised downstream region (Korb et al., 2012; Smetacek et al., 2004). Phytoplankton blooms at P3 can be sustained for 3-4 months (Whitehouse et al., 2008), whereas blooms are typically much shorter in the SACCF region where P2 is located (Park et al., 2010), likely influencing the dynamics of the zooplankton community. Variability in regional dispersal or retention by the current systems of the ACC is important for determining the seasonal dynamics of Scotia Sea ecosystems (Murphy et al., 2007; Thorpe et al., 2007)."

As now more fully described in the methods (lines 101-134), current meter data at the study sites, suggest that the currents in the vicinity of the ST are low (mean current velocities were <7.2cm s$^{-1}$ and <14.2cm s$^{-1}$ at P3 and P2 in 2013) and so we would not expect a large amount of horizontal drift of zooplankton species at depth.

83ff: I struggled a bit with the different locations abbreviations (D1/ D2/ P2/ P3) and would suggest sticking to one label (e.g. P2/ P3 as also marked in the map).

AC: Thank you for this recommendation, we have removed reference to D1 and D2 and also H2009 and H2010 which referred to the historical data. These abbreviations have not been used in the text and we have amended our description of the use of the 2009 and 2010 data from Manno et al., 2015 as follows for clarity (lines 238-242):

"As only an average FP size for each morphological type (rather than for all individual FP) was measured for samples from the ST deployments, we make use of historical sediment trap data (Manno et al., 2015) at the same sites from December 2009 and 2010. The size of all FP in each sample-split were measured in the study of Manno et al. (2015) and hence we use these data to compare size distributions of MSC and ST collected FP."

85-89: Consider moving this information to the paragraph "sediment trap deployment". I think this would be a more natural place to have it

AC: Moved as suggested to: 2.3.2. sediment trap deployments.

93ff: Where have the nets been taken?

AC: Thank you for spotting this, we have added in the information of where the Bongo nets were deployed (lines 137-138):

"Mesozooplankton samples were collected at both P2 and P3 using a motion-compensating Bongo net (61 cm mouth diameter, 2.8 m long, 200 μm mesh)."

111: literature values – did you have similar zooplankton species as in Stamieszkin et al (2015) that you can assume that the estimated made are meaningful?

The Stamieszkin et al. (2015) paper focuses on species found in the Gulf of Maine in the North Atlantic, yet a number of species occur in common with our study, including *Metridia lucens, Oithona spp., Clausocalanus spp., Pleuromamma robusta,* and *Candacia spp.* In addition the size range of prosome length (PL) of the species in Stamieszkin et al., (2015) is 0.025 – 5.6 mm (mean 1.1 mm, median 0.88mm) which is similar to, but slightly smaller than, the range of PL in our study 0.17 -7.2 mm (mean 2.1 mm, median 1.5 mm). We therefore believe that using the Stamieszkin et al. (2015) relationship provides a valid estimate for our study. In addition, similar relationships relating copepod PL to FP volume were found by Uye and Kaname (1994) using data

collected in the Inland Sea of Japan, as too by Mauchline (1998, see pg. 198) based on a review of previous literature. Thus suggesting that these relationships are more generally applicable.

144: How did you do the classification of the FP? I totally understand that you get a feeling for it, but was it more a subjectively choice (eye-balling) or did you e.g. take pictures and use roundness/ ratio of the minor/major axis? This is more a question out of curiosity, but I think that some standards should be established to make FP classification as objective as possible.

AC: We completely agree that standards should be established as this would help make literature studies more comparable. The size classification was based on the classifications of Manno et al. (2015) as we found FP of the same characteristics in the marine snow catcher samples. This was based on visual assessment using light microscopy having inspected all the samples to identify the main FP classes. We did not use any ratios of the major axis, in part because this can be difficult if FP are fragmented. We agree that it would be good to begin to populate data on the dimensions of FP, shape characteristics and where possible the known producer, as this would help future studies of sinking FP. It is difficult to assign a particular producer to a FP even when the size and shape are known as size in particular can vary both with and between stages of a particular species, and may change seasonally with factors such as diet, as is seen for carbon contents and egestion rates (Atkinson et al., 2012; Manno et al., 2015).

Results: 251-253: This sentence confuses me... Please revise the language.

AC: We have restructured the sentence and quoted values from Table 2 to improve the clarity of this sentence (lines 316-319).

"At P3, the flux of cylindrical and elliptical FP in the MSC was an order of magnitude higher than fluxes of round or ovoid FP (190,716 FP $m^{-2}$ $d^{-1}$ compared to 32,172 FP $m^{-2}$ $d^{-1}$). Similarly at P2, cylindrical and elliptical FP were the dominant FP type (21,128 FP $m^{-2}$ $d^{-1}$), but fluxes of round FP were also important (14,596 FP $m^{-2}$ $d^{-1}$) at this site (Table 2). FP fluxes in the ST were dominated by ovoid FP at both sites (Table 2)."

Discussion: 255: mainly copepod? See general comments

AC: Changed to copepod community to reflect that we focus on these species.

277-282: Good and important point!

362-365: Please revise language to make easier understandable

AC: We have amended these sentences (Lines 469-487) as shown below:

"Zooplankton are more concentrated in the epipelagic, however, the total abundance of zooplankton in the meso- and bathypelagic can be high due to the large depth extent of these layers. In the Antarctic Zone (to the west of our study site), Ward et al. (2014) found that the total depth integrated zooplankton abundance in the 250-2000 m horizon (extrapolating abundances recorded at 750-1000 m down to 2000 m) is about three quarters (0.74) of the zooplankton abundance in the top 250 m. Therefore it is likely that there is still substantial production of FP in the lower mesopelagic, and compared to FP produced in the epipelagic, FP produced in the lower mesopelagic are subject to remineralisation processes over a shorter distance so are more likely to reach the deep ocean intact."

379: "the increased abundance of small copepods at P2": I am unsure what you mean by the term "increased abundance" (increased compared to what?) and Fig. 2 shows that there were more small copepods at P3 compared to P2. Thus, I do not understand the argument made in line 379-380.

AC: We have altered this section (Lines 498-501) to explain our argument more clearly.

"Regardless of the mechanism, previous studies agree that high microcopepod abundances can lead to increased FP retention. The ratio of small copepods to large calanoids is higher at P2 (Fig. 2), which may result in greater losses of FP in the epi- and mesopelagic, resulting in lower numbers of FP captured in our MSC and ST at P2. Indeed, we see higher attenuation of FP fluxes at P2 than P3 between our measurement depths (Table 2)."

383-387: Do you have any data that you have a larger deep ZP community at P3 than at P2? Otherwise the argumentation is a bit thin.

AC: Unfortunately we do not have sufficient data to provide strong evidence for this, so have removed these sentences to avoid speculation.

387-392: Can you please make your arguments a bit clearer? I have problems under- standing what you mean here.

AC: The sentences have been restructured as follows (lines 504-510) to make our argument clearer:

"Our data implies that in situ FP production in the mesopelagic accounted for additional fluxes of FP to the bathypelagic at both P2 and P3. However as there is the potential for further working, fragmentation and remineralisation of FP produced in the mesopelagic, the gross deep FP production cannot be quantified here. We therefore cannot determine whether higher FP fluxes at P3 are due primarily to reduced FP attenuation or to increased FP production at depth; most likely a combination of both mechanisms is taking place. Previous work in the region, has however found that in the upper mesopelagic (mixed layer depth-200 m) FP attenuation is higher at P2 than P3 (Belcher et al., 2016)."

392: Consider replacing the sentence by something like: "Most likely a combination of both mechanisms takes place."

AC: Changed as suggested

396ff: Please be a bit more cautions in your conclusions as you were previously in the discussion (line 351).

AC: We have changed the wording of our sentences so as not to overstate our findings in acknowledgement of the uncertainties, hence we do not specify that vertical migration must be occurring (lines 512-516).

"Our comparison of FP size, shape and abundance in the upper mesopelagic and lower bathypelagic agrees with previous hypotheses (Accornero et al., 2003; Manno et al., 2015; Suzuki et al., 2003), that in situ FP production augments the flux of FP to depth in the Southern Ocean. We find that the occurrence of intact FP in deep ST could be explained by both vertical migrations of zooplankton, and repackaging and in situ FP production by meso- and bathypelagic zooplankton populations (Fig. 7)."

398-412: Please revise to make it easier for your reader to follow (e.g. 400: deep ocean – what do you mean by that; 402: depth of migration – how deep it that?) and also revise your Fig. 7 to interlink text and figure better).

AC: We have revised this section to make our calculations clearer. This section serves to highlight that the route by which FP are transferred through the mesopelagic plays an important control on the amount of carbon that is transferred. See lines 514-572.

"We find that the occurrence of intact FP in deep ST could be explained by both vertical migrations of zooplankton, and repackaging and in situ FP production by meso- and bathypelagic zooplankton populations (Fig. 7). Taking an integrated surface production of 1 g C $m^{-2}$ $d^{-1}$ (based on measurements by Korb et al. (2012) to the northwest of South Georgia), and assuming an assimilation efficiency of 66% (Anderson and Tang, 2010; Head, 1992) during vertical migration (left panel Fig. 7, Case A), we calculate that up to 340 mg C $m^{-2}$ $d^{-1}$ could reach the depth of migration (this depth will vary both between species and seasonally). In comparison, if FP are repackaged multiple times on their transit through the mesopelagic then FP will be assimilated multiple times, resulting in reduced transfer of carbon when compared to diel vertical migration. For example, FP that are assimilated twice over the same vertical distance as a typical vertical migration (right panel, Fig. 7, Scenario 2), result in up to 115 mg C $m^{-2}$ $d^{-1}$ reaching the same depth. The exact difference in carbon transfer between these two routes (Case A and B) will depend on the number of repackaging steps over the migration depth, specific assimilation efficiencies of the repackaging copepods as well as loss of FP carbon via remineralisation. However, these calculations highlight that the route by which the FP are transferred to depth is a key control on the amount of carbon reaching depth. Regardless of the feeding mode of these mesopelagic zooplankton communities (detritivory, omnivory or carnivory), production of FP at depth via both the aforementioned scenarios supports the transfer of intact FP to the deep ocean, supporting the sequestration of carbon on long timescales."

Additionally, we have changed figure 7 in line with your suggestions, see detailed response in general comments.

Figures and Tables:

Figure 1: You do not mention the different fronts in the text. Is there any reason to include them here? In my opinion, depth lines could be more interesting
– perhaps that could also help to explain the stronger attenuation of FP at P2?

AC: We choose to keep the fronts in Figure 1 as these are now mentioned in the text in the Study Site section, and we think they provide useful information on the environmental setting of the two sites P2 and P3, highlighting clearly their upstream and downstream locations. The shading in the plot displays the bathymetry; we have added a scale bar to illustrate the depths in the region.

Figure 7: See general comment

AC: See detailed response in general comments

Technical corrections:
205: change to "<2 mm" to be consistent (no space after the <)
AC: Amended
205: change "is similar" to "was similar"
AC: Amended
192: past tense? Change to "we took into account"?
AC: Amended

Figure 2: Other zooplankton (see text for full information…) – I could not find that information

AC: The species comprising the small and large copepod groups are detailed in the results section (lines 263-266). Additionally we have added the following text to the methods (lines 148-151):

"Zooplankton were grouped into; small microcopepod species (*Oithona similis*, *Oncaea sp.* and *Ctenocalanus sp.*) large calanoid copepod species (*Rhincalanus gigas, Calanoides acutus, Calanus similimus, C. propinquus, Euchaeta spp.,* and *Metridia spp*), small euphausiids (all euphausiid species caught in net) and other zooplankton (all remaining species)."

Figure 3/ 5 / 6: Unsure about the labelling of the x-axis. Why do you have the step from 0.005 to 0.008?

AC: Uneven size classes have been used to best represent the large spread in FP volumes. As we estimate that the copepod community produced high numbers of FP the small size classes, we were able to resolve size differences at a higher resolution than for larger FP which were produced in smaller numbers. We keep the same sizes classes for MSC and ST samples to allow easy comparison.

Table 2: Minor aspect, but is there a reason, why you put P2 on the right side of the table and P3 on the left side? Intuitively, I would do expect it the other way round (as in Fig. 2)

AC: Good suggestion, we have swapped the columns around.
* * *
References cited in response to review:

[revised manuscript text omitted]

---

## Author Comment (AC2) · 22 Feb 2017

**Response to reviewer #2**

AC: Thank you for reviewing our manuscript and for taking the time to make suggestions for improvement, please find our responses below. Line numbers refer to the marked up version of the manuscript which has also been uploaded.

**Anonymous Referee #2**

"Zooplankton fecal pellet transfer through the meso- and bathypelagic layers in the Southern Ocean in spring" by Belcher, A., Manno, C., Ward, P., Henson, S., Sanders, R., Tarling, G.

General comments
The manuscript by Belcher et al. compares estimates of copepod faecal pellet production in the epipelagic with those of faecal pellet abundances in the meso- and bathypelagial derived from Marine Snow Catcher (MSC) and sediment traps. The study was conducted in the Scotia Sea, Southern Ocean. Based on faecal pellet morphology and abundance, the main conclusion of this study is that small faecal pellets are in high abundance in the epipelagic but do not contribute much to export fluxes. Instead, repackaging of faecal pellets and de novo production take place in the meso- and bathypelagic. The manuscript is well written. The findings are in accordance with expectations derived from earlier studies on faecal pellet export in the ocean.

Specific comments
I suggest that the authors provide more information on their methods and measurements to allow for better evaluation of the data. I also suggest a more critical discussion on the comparability of data on faecal pellet fluxes derived from net tows, MSC and sediment traps.
1) Lines 89 and 90; here, an estimate for a mean current velocity at the study site of <10 m s-1 is given and Whitehouse et al. (2012) are cited to suggest that lateral advection can be neglected. It is not clear where the current velocity was measured and for what processes it can be neglected. I assume that a current meter was used in the cited study to estimate the trapping efficiency of the sediment traps, but this information should be given. Is the current velocity in the epi- and mesopelagic in the Scotia Sea in the same range? Does one have to consider lateral advection of faecal pellets in the water column? This is important to estimate how well samples from a MSC and deep traps can be compared.

AC: Thank you for drawing our attention to this, we realise that there was a typo and that mean velocities at the site are < 10 cm s$^{-1}$. This is based on the work of Whitehouse et al. 2012 for the study area, consistent with Manno et al. 2015 whose sediment trap data we have used for our comparisons. We additionally present current meter data from the P2 and P3 moorings collected between 2012 and 2014 (lines 101-134). Unfortunately we do not have current meter data from the mesopelagic to for comparison. However, Heywood and King, (2002) measured, geostrophic velocities in the upper 1000 m of Scotia Sea of <10 cm s$^{-1}$ suggesting currents here are in the same range. Considering that zooplankton FP sink rapidly and that vertical shear is small below the mixed layer of the ACC (Firing et al., 2011), we do not think lateral advection will have significantly biased our results. See also response to point 4 of reviewer 2 where we add additional discussion on the uncertainties of the spatial and temporal resolutions of the different sampling techniques.

"Mean current velocities in December 2012 and 2013 (measured with a Nortek Aquadopp current meter deployed just below the ST) were 7.2 and 4.5 cm s$^{-1}$, and, 14.2 and 12.5 cm s$^{-1}$ at P3 and P2 respectively. These data agree with mean current velocities at the depth of the ST at both sites of <10 cm s$^{-1}$ observed by Whitehouse et al., (2012) in 2008, suggesting that the effects of lateral advection are minimal and as such they are not considered in this study."

2) 140ff.; please indicate how many splits were analyzed. Were the splits analyzed separately so that a sampling error can be given? How many pellets per split, or in total, were counted? So far only relative abundances are given in the manuscript and supplemental information. The authors may consider providing a table with absolute counts in the supplemental information. Please give this information also for faecal pellets determined in the MSC samples.

AC: We have added additional text to clarify our methods in terms of splits analysed.

Lines 177-178:

"All particles collected in the MSC tray were counted as it was not necessary to split the sample."

Lines 197-198

"Three replicates were analysed for ST FP, with all FP in each replicate counted (see supplementary table S1 for absolute counts)."

For the sediment trap samples splits were 1/16 and 1/160 of the total sample size for P2 and P3 respectively. Three replicate splits were calculated and following your advice we have detailed the mean and standard deviation of these counts in supplementary table S1. We have also included the absolute counts of FP collected in the MSC. This is the total number of FP in the MSC as it was not necessary to split the sample. We are aware that the sample size for the MSC is small at the P2 site which is controlled by the particle flux at the time of sampling. However, data from multiple deployments over two field campaigns increases our confidence in these values. These data are the best estimations available for comparison with the ST at our study sites and we believe they still provide insights into the transfer of FP through the mesopelagic.

3) 159 ff.: The quality of the faecal pellet sinking velocity measurement and therewith of the faecal pellet fluxes cannot be evaluated. The authors state that they used two different approaches to determine faecal pellet sinking velocity and that there were no significant differences between the methods. But how reliable are the obtained sinking velocities? The range of sinking velocities given in line 246 is rather large (24- 950 m d$^{-1}$). The ranges given in lines 248-249 are much smaller. How do these numbers compare? What was the variability of sinking velocity within each approach? What was the variability within each size class? Since these data are used to calculate the faecal pellet flux (FPF), the original data should be given in the manuscript and their accuracy assessed critically. Please add more information.

AC: We have amended section 3.4 to include only the sinking velocities of non-krill FP as these are the data utilised in this study, and the high sinking velocity (950 m d$^{-1}$) of one particularly large krill FP is the cause of the large range that you mention. The text now reads as follows (lines 311-314):

"Sinking velocities of FP (excluding krill FP) collected in the MSC ranged from 52 to 382 m d$^{-1}$ at P2 and 13 to 227 m d$^{-1}$ at P3 reflecting the range in FP shapes and sizes. Generally small FP had lower sinking velocities than larger FP. We measured FP sinking rates (excluding krill FP) of 47-120 m d$^{-1}$ for FP <0.002 mm$^3$, and 36-270 m d$^{-1}$ for FP >0.02 mm$^3$ (supplementary table S2). Rates measured in this study are consistent with the range of 5-220 m d$^{-1}$ given by Turner (2002) for copepod FP."

Both techniques used in this study have been used to measure sinking velocities of FP and other sinking particles in other published studies, e.g. Cavan et al. (2015), Iversen and Ploug (2013), Iversen et al. (2010), Ploug et al. (2008), Riley et al. (2012) and Small et al. (1979), and hence we believe that these methods are justified in their use in this study. There are inevitably limitations when carrying out sinking velocity measurements in the field, but care was taken to ensure that measurements were not made in rough conditions when significant movement of the research vessel could have impacted results. We have included the sinking velocity data measured in this study in supplementary data S2 for transparency to show clearly the range in sinking velocities measured. This range in sinking velocities is not unexpected considering the range in size

and type of FP that we observed and is consistent with previous literature (review of Turner (2002) gives a range of 5-220 m d$^{-1}$ for copepod FP).

4) The authors conclude that small FP that sink more slowly are not transferred efficiently to depth as they are subject to remineralization and coprophagy for a much longer period of time than fast sinking large particles. This is very well comprehensible. However, I would like to see a more critical discussion of the comparability of data from net tows, MSC and sediment traps, which takes into account the spatio-temporal variability in that region, the three-dimensional flow field and the current velocities. In addition to biogenic loss process, slowly sinking fecal pellets found in the MSC of stations P 2 and P3 may not be represented well in the sediment traps at 1500 and 2000m depth, because the traps collect particles from a much wider area (e.g. Waniek et al. 2000) and integrate over a longer, and different, time period.

AC: The differences in the temporal and spatial resolutions of the three sampling methods is indeed an important consideration. We have followed your recommendation and added the following text (Lines 458-451) to discuss the various scales and acknowledge the uncertainties.

"When comparing datasets collected via different methods (in this case Bongo nets, MSC and ST), it is important to consider the different time and space scales over which they measure. The zooplankton Bongo net samples integrated vertically over the top 200 m and temporally over the period over which replicate samples were taken (a few days at each site for both cruises). MSC samples were an instantaneous snapshot of the particle flux and, at a deployment depth of 110 m below the mixed layer, they integrate over spatial scales of tens of kilometres (based on median sinking rates at P2 and P3 and a current speed of 10 cm s$^{-1}$). Conversely ST samples captured the flux over a 15 day period and at a deployment depth of 1500 and 2000 m had a potential sample collection area on spatial scales of hundreds of kilometres (based on the same conditions). If zooplankton communities vary significantly over tens of kilometres then this would reduce the direct comparability of MSC and ST data. Previous studies in the region suggest that much of the Scotia Sea is populated by a single zooplankton 'community', but there are regional differences in the stage of phenological development. (Ward et al., 2006), implying that the species composition may not vary on short spatial scales. Changes in the species stage are likely tied to changes in phytoplankton productivity, as for much of the time, Southern Ocean zooplankton are food limited (Ward et al., 2006). Cluster analysis of phytoplankton in the Scotia Sea reveals distinct communities (in terms of abundance, community structure and productivity) on spatial scales of hundreds of kilometres (Korb et al., 2012), and hence we would not expect significant changes in the stage-structure of zooplankton on the spatial resolution of the MSC, making these results more comparable to those of the ST. The high sinking rates of zooplankton FP means that their occurrence in ST is representative of the conditions directly above the ST (Buesseler et al., 2007). Slow-sinking particles spread out more as they sink which increases our uncertainty in depth comparisons of smaller FP. However, the spatial scale of zooplankton variability at our study site means that slow-sinking FP particles reaching the ST likely reflect the same zooplankton community structure as occurring directly above the ST. For each of our three methods (nets, MSC and ST), we take averages over multiple years which should also reduce the uncertainties associated with the various spatial and temporal resolutions of the three methods. However, we acknowledge that the different spatial and temporal scales of measurement could also contribute to some of the vertical changes in FP shape and size structure that we observed."

References cited in response to review:

Buesseler, K. O., Antia, A. N., Chen, M., Fowler, S. W., Gardner, W. D., Gustafsson, O., Harada, K., Michaels, A. F., Rutgers van der Loeff, M., Sarin, M., Steinberg, D. K. and Trull, T.: An assessment of the use of sediment traps for estimating upper ocean particle fluxes, J. Mar. Res., 65(3), 345–416, doi:10.1357/002224007781567621, 2007.

Cavan, E. L., Le Moigne, F., Poulton, A. J., Tarling, G. A., Ward, P., Daniels, C. J., G, F. and Sanders, R. J.: Attenuation of particulate organic carbon flux in the Scotia Sea, Southern Ocean, controlled by zooplankton fecal pellets, Geophys. Res. Lett., 42(3), 821–830, doi:10.1002/2014GL062744, 2015.

Firing, Y. L., Chereskin, T. K. and Mazloff, M. R.: Vertical structure and transport of the Antarctic Circumpolar Current in Drake Passage from direct velocity observations, J. Geophys. Res., 116, doi:10.1029/2011JC006999, 2011.

Heywood, K. J. and King, B. A.: Water masses and baroclinic transports in the South Atlantic and Southern oceans, J. Mar. Res., 60(5), 639–676, 2002.

Iversen, M. H. and Ploug, H.: Temperature effects on carbon-specific respiration rate and sinking velocity of diatom aggregates - potential implications for deep ocean export processes, Biogeosciences, 10, 4073–4085, doi:10.5194/bg-10-4073-2013, 2013.

Iversen, M. H., Nowald, N., Ploug, H., Jackson, G. A. and Fischer, G.: High resolution profiles of vertical particulate organic matter export off Cape Blanc, Mauritania: Degradation processes and ballasting effects, Deep Sea Res. Part I Oceanogr. Res. Pap., 57(6), 771–784, doi:10.1016/j.dsr.2010.03.007, 2010.

Korb, R. E., Whitehouse, M. J., Ward, P., Gordon, M., Venables, H. J. and Poulton, A. J.: Regional and seasonal differences in microplankton biomass, productivity, and structure across the Scotia Sea: Implications for the export of biogenic carbon, Deep Sea Res. Part II Top. Stud. Oceanogr., 59–60, 67–77, doi:10.1016/j.dsr2.2011.06.006, 2012.

Ploug, H., Iversen, M. H., Koski, M. and Buitenhuis, E. T.: Production, oxygen respiration rates, and sinking velocity of copepod fecal pellets: Direct measurements of ballasting by opal and calcite, Limnol. Oceanogr., 53(2), 469–476, 2008.

Riley, J. S., Sanders, R., Marsay, C., Le Moigne, F., Achterberg, E. P. and Poulton, A. J.: The relative contribution of fast and slow sinking particles to ocean carbon export, Global Biogeochem. Cycles, 26, doi:10.1029/2011GB004085, 2012.

Small, L. F., Fowler, S. W. and Ünlü, M. Y.: Sinking rates of natural copepod fecal pellets, Mar. Biol., 51(3), 233–241, doi:10.1007/BF00386803, 1979.

Turner, J. T.: Zooplankton fecal pellets, marine snow and sinking phytoplankton blooms, Aquat. Microb. Ecol., 27, 57–102, doi:doi:10.3354/ame027057, 2002.

Ward, P., Shreeve, R., Atkinson, A., Korb, B., Whitehouse, M., Thorpe, S., Pond, D. and Cunningham, N.: Plankton community structure and variability in the Scotia Sea: austral summer 2003, Mar. Ecol. Prog. Ser., 309, 75–91, doi:10.3354/meps309075, 2006.

Whitehouse, M. J., Atkinson, A., Korb, R. E., Venables, H. J., Pond, D. W. and Gordon, M.: Substantial primary production in the land-remote region of the central and northern Scotia Sea, Deep Sea Res. Part II Top. Stud. Oceanogr., 59–60, 47–56, doi:10.1016/j.dsr2.2011.05.010, 2012.

---

## Author Comment (AC3) · 22 Feb 2017

National Oceanography Centre

European Way

Southampton SO14 3ZH

United Kingdom

22/02/2017

Copernicus Gesellschaft mbH

Bahnhofsallee 1e

37081 Göttingen

Germany

**Re: BG – 2016-520 Belcher et al.**

Dear Dr Gerhard Herndl,

Please find enclosed our revised manuscript (marked up version), complete with tables, figures and supplementary material. We submit this in response to the comments from two anonymous reviewers and have addressed their concerns. Please see also our point-by-point response to each reviewer which has been uploaded on the Biogeosciences Discussion page.

In particular, we have revised the manuscript title and text to make sure it reflects our focus on copepods rather than the whole zooplankton community. Additionally, we have examined current meter data to back up our statement that lateral advection did not play an important role at the depth of the sediment traps. We also now provide the absolute counts of faecal pellets in both sediment trap and marine snow catcher samples, as well as faecal pellet sinking velocities. We hope this provides transparency to our work and also provides useful data for future studies.

We think that the manuscript has been improved and thank the two anonymous reviewers for their helpful suggestions. Thank you for taking the time to review this submission, we look forward to hearing from you.

Yours Sincerely,

Anna Belcher

[revised manuscript text omitted]

    **Supplementary Table**

**Table S1: Absolute number of FP counted in sediment trap (ST) sample split and Marine Snow Catcher (MSC) samples. Three**
**replicates were counted for ST samples and are presented as mean (standard deviation), where as all FP collected in the MSC**
**samples were counted. Krill FP are not included.**

| Cruise | Site | Sampling Method | # FP |
|--------|------|-----------------|------|
| JR291 | P2 | MSC | 4 |
| | P2 | MSC | 9 |
| | P2 | MSC | 28 |
| | P3 | MSC | 15 |
| | P3 | MSC | 74 |
| JR304 | P3 | MSC | 120 |
| | P3 | MSC | 252 |
| | | | |
| Dec 2009 | P2 | ST | 422 (98) |
| | P3 | ST | 1156 (195) |
| Dec 2010 | P2 | ST | 564 (134) |
| | P3 | ST | 974 (238) |

**Table S2: Sinking velocities and volumes of FP (excluding krill FP) collected in Marine Snow Catchers at P2 and P3 during**
**research cruises JR291 and JR304.**

| Site | FP volume (mm³) | FP sinking velocity (m d⁻¹) | Site | FP volume (mm³) | FP sinking velocity (m d⁻¹) |
|------|-----------------|------------------------------|------|-----------------|------------------------------|
| P2 | 0.040 | 144 | P3 | 0.010 | 75 |
| P2 | 0.031 | 270 | P3 | 0.027 | 57 |
| P2 | 0.008 | 52 | P3 | 0.002 | 48 |
| P2 | 0.040 | 144 | P3 | 0.026 | 87 |
| P2 | 0.031 | 135 | P3 | 0.002 | 51 |
| P2 | 0.057 | 134 | P3 | 0.005 | 68 |
| P2 | 0.019 | 342 | P3 | 0.014 | 49 |
| P2 | 0.011 | 382 | P3 | 0.028 | 92 |
| P2 | 0.072 | 247 | P3 | 0.023 | 106 |
| P2 | 0.044 | 101 | P3 | 0.009 | 24 |
| P2 | 0.007 | 193 | P3 | 0.091 | 92 |
| P2 | 0.017 | 116 | P3 | 0.066 | 140 |
| P2 | 0.035 | 207 | P3 | 0.012 | 57 |
| P2 | 0.002 | 246 | P3 | 0.006 | 65 |
| P2 | 0.016 | 61 | P3 | 0.010 | 62 |
| P2 | 0.001 | 120 | P3 | 0.006 | 64 |

| | | | | | |
|---|---|---|---|---|---|
| P2 | 0.003 | 98 | P3 | 0.002 | 47 |
| | | | P3 | 0.037 | 36 |
| | | | P3 | 0.031 | 53 |
| | | | P3 | 0.014 | 122 |
| | | | P3 | 0.021 | 36 |
| | | | P3 | 0.077 | 100 |
| | | | P3 | 0.018 | 62 |
| | | | P3 | 0.026 | 64 |
| | | | P3 | 0.013 | 79 |
| | | | P3 | 0.083 | 227 |
| | | | P3 | 0.286 | 203 |
| | | | P3 | 0.165 | 189 |
| | | | P3 | 0.007 | 100 |
| | | | P3 | 0.006 | 74 |
| | | | P3 | 0.005 | 13 |
| | | | P3 | 0.115 | 106 |
| | | | P3 | 0.021 | 60 |
| | | | P3 | 0.005 | 68 |
| | | | P3 | 0.018 | 79 |
| | | | P3 | 0.006 | 49 |
| | | | P3 | 0.009 | 64 |
| | | | P3 | 0.003 | 155 |
| | | | P3 | 0.005 | 222 |
| | | | P3 | 0.256 | 144 |
| | | | P3 | 0.002 | 82 |
| | | | P3 | 0.006 | 133 |

---

## Author Response (AR1)

National Oceanography Centre

European Way

Southampton SO14 3ZH

United Kingdom

03/03/2017

Copernicus Gesellschaft mbH

Bahnhofsallee 1e

37081 Göttingen

Germany

**Re: BG – 2016-520 Belcher et al.**

Dear Dr Gerhard Herndl,

Thank you for taking the time to review our revisions to the manuscript and for approving our changes. We have uploaded the manuscript as requested with no further changes.

Yours Sincerely,

Anna Belcher

[revised manuscript text omitted]

**Supplementary Material**

**Table S1: Absolute number of FP counted in sediment trap (ST) sample split and Marine Snow Catcher (MSC) samples. Three**
**replicates were counted for ST samples and are presented as mean (standard deviation), where as all FP collected in the MSC**
**samples were counted. Krill FP are not included.**

[revised manuscript text omitted]

      **Supplementary Figures**

[Figure]

[Figure]

**Figure S1: Comparison of sediment trap faecal pellet (FP) morphologies measured in this study (2013 and 2014, black) with those**
**measured historically (2009 and 2010, white) at a) P2 and b) P3. Both studies are means of November and December data. The**
**percent (%) of FP in each category is broadly consistent between study years (paired t-test p>0.5) providing support for our use of**
**historical data for size comparisons with marine snow catcher data collected in 2013 and 2014.**

[Figure]

[Figure]

**Figure S2: Mesozooplankton abundances in the Scotia Sea. Average (±SE) abundance (ind. m⁻²) from Bongo net tows (0-200 m,**
**200 μm mesh) taken during cruises JR291 and JR304 for a) P2 and b) P3. Note the log scale on the y axis.**